# Multi-dimensional scaling techniques unveiled gain1q&loss13q co-occurrence in Multiple Myeloma patients with specific genomic, transcriptional and adverse clinical features

Carolina Terragna [1,3] ✉, Andrea Poletti [1,2,3], Vincenza Solli[1,2], Marina Martello [1,2], Elena Zamagni[1,2], Lucia Pantani[1], Enrica Borsi[1], Ilaria Vigliotta[1,2], Gaia Mazzocchetti[1,2], Silvia Armuzzi[1,2], Barbara Taurisano[1,2], Nicoletta Testoni[1,2], Giulia Marzocchi[1,2], Ajsi Kanapari[1,2], Ignazia Pistis[1], Paola Tacchetti[1], Katia Mancuso[1,2], Serena Rocchi[1,2], Ilaria Rizzello [1,2] & Michele Cavo[1,2]

The complexity of Multiple Myeloma (MM) is driven by several genomic aberrations, interacting with disease-related and/or -unrelated factors and conditioning patients' clinical outcome. Patient's prognosis is hardly predictable, as commonly employed MM risk models do not precisely partition high- from low-risk patients, preventing the reliable recognition of early relapsing/refractory patients. By a dimensionality reduction approach, here we dissect the genomic landscape of a large cohort of newly diagnosed MM patients, modelling all the possible interactions between any MM chromosomal alterations. We highlight the presence of a distinguished cluster of patients in the low-dimensionality space, with unfavorable clinical behavior, whose biology was driven by the co-occurrence of chromosomes 1q CN gain and 13 CN loss. Presence or absence of these alterations define MM patients overexpressing either *CCND2* or *CCND1*, fostering the implementation of biology-based patients' classification models to describe the different MM clinical behaviors.

Despite the availability of new drugs and of effective therapeutic protocols, Multiple Myeloma (MM) is still a hard-to-treat haematologic cancer[1]. Indeed, even though the therapy response rates and the patients' survival have overall greatly improved, a substantial proportion of patients still do not have an actual benefit in terms of disease-free survival. To further improve the outcome of these patients, the identification of discrete biological entities of the disease, strictly defined by specific genomic features, is crucial. This would foster their stratification based on the underlying biology of the tumour, eventually supporting a tailored therapy, targeted on patients'-specific vulnerabilities[2–4].

Currently, there are two main recognized genetic events that cause MM: odd-numbered chromosome trisomies (hyperdiploidy, HD) and chromosome 14q32 immunoglobulin heavy chain locus

[1]IRCCS Azienda Ospedaliero-Universitaria di Bologna—Istituto di Ematologia "Seràgnoli", Bologna, Italy. [2]DIMEC—Department of Medical and Surgical Science, University of Bologna, Bologna, Italy. [3]These authors contributed equally: Carolina Terragna, Andrea Poletti. ✉e-mail: carolina.terragna@unibo.it

**Fig. 1 | Violin plots of CNAs landscape.** Overview of all CNAs observed in (**a**) BO dataset and (**b**) CoMMpass dataset. Each dot represents a single CN event. Events are grouped by chromosome arm. Red dots represent "clonal" alterations (affecting virtually all, i.e., >90%, tumour cells), green dots represent "sub-clonal-major" alterations (affecting the majority, i.e., between 50% and 90%, of tumour cells); blue dots represent "sub-clonal-minor" alterations (affecting the minority, i.e., between 10% and 50%, of tumour cells). **c, d** Venn diagrams illustrating the common genomic ground of chromosomal aberrations in BO dataset and CoMMpass dataset. The presence of at least one out of the four most common genomic lesions, i.e., gain 1q, loss 13q, HD and t-IgH, was sufficient to describe the totality of patients.

translocations (t-IgH)[5–7]. However, newer technologies have shown that other genetic changes also play a role in MM, meaning that MM patients cannot be simply divided into two categories based on these two events. In addition, some patients don't present either of these events, and some present both[5].

Recently, the integration of data coming from genome-wide and transcriptome analyses, aimed at the definition of the conditional dependencies of MM driver events, showed that discrete subgroups of patients could be defined, whose genomic profile included several co-operating, nonrandomly distributed events[8–10]. Notably, these studies highlight that the strongest determinant of the genomic substructure of these subgroups of patients remained t-IgH and Copy Number Alterations (CNAs)[8,10], even when mutation and gene fusions data are included in group clustering. Indeed, the final MM groupings defined in those papers are defined mostly by CNA and t-IgH.

Despite the novelty of this new disease perspective, the clinical implementation of MM patients' stratification is conditioned by the complexity of the technical approach needed to define the subgroups and by the observation that a reliable prediction of MM clinical course remains elusive[11–14]. In fact, the need for one univocal MM scoring system, fitting all clinical circumstances, is barely met to date[15,16].

A possible reason might be ascribed to the genomic and therapeutic heterogeneity of MM, which should be considered as a complex system. Therefore, the use of cytogenetic aberrations as mere independent factors to be included in statistical models, rather than as the expression of specific biological damages, might be the cause of the traditional, narrow-sighted genomic picture of MM, not clinically exploitable.

In the present paper, we aim at describing the landscape of whole-genome CNAs and t-IgH in a large cohort of newly diagnosed MM patients, up-front treated with standard-of-care regimens (including Proteasome Inhibitors, PIs and Immunomodulators, IMiDs), by using a dimensionality reduction statistical approach. We define and characterize a sub-group of MM patients with unfavourable clinical behaviour, whose biology is driven either by the co-occurrence of chromosome 1q CN gain and chromosome 13 CN loss or by the absence of both alterations. We show that the presence of these alterations defines MM patients overexpressing *CCND2* and, on the contrary, their absence defines MM patients overexpressing *CCND1*, thus mirroring the previously reported mutually exclusive pattern of expression of cyclin D genes, proven as early and unifying event in MM pathogenesis[17].

## Results

### Multiple myeloma genomic landscape

The genomic landscape of 513 newly diagnosed MM (NDMM) patients (BO-dataset) was deeply explored both by SNPs array, to assess CNAs profile, and by FISH, to detect t-IgH. In parallel, the genomic landscape of 840 NDMM patients obtained from the CoMMpass Multiple Myeloma Research Foundation study IA14 was also analysed: here, CNAs and translocations data were obtained by whole genome sequencing (WGS) data. We validated our findings using this independent data set generated using a different detection technology (Next Generation Sequencing, NGS, which also provided an opportunity to test the robustness of our results).

CNA events were identified at the chromosome arm level, which is the gold standard for CNA detection as per FISH, and annotated when either broad (spanning >25% of the chromosome arm) or covering any one of the focal genomic regions described in Supplementary Fig. S1. CNAs were widely distributed along all chromosomes and were defined as clonal (i.e., equally affecting all, or >90%, of evaluated plasma cells), sub-clonal "major" (i.e., affecting most, or >50%, of evaluated plasma cells), or sub-clonal "minor" (i.e., affecting a minority, between 10% and 50%, of evaluated plasma cells) (Fig. 1). In order to exclude potential technical biases generated by signal noise and to

focus on tumour-characterizing alterations, we chose to consider both clonal and sub-clonal "major" events, excluding from the present analysis all sub-clonal "minor" events.

The whole catalogue of observed CNAs is described in Fig. 1 and reported in Supplementary Data S1.

By ranking all the considered CNAs according to their frequencies, three lesions resulted more recurrent, as compared to others: CN losses on chromosome 13q (loss 13q 225/513, 44%), odd-numbered chromosomes CN gains (HD 210/513, 41%), CN gains on chromosome 1q (gain 1q, 157/512, 31%). Similarly, the sub-clonal CNAs overall recurrence was lower, with CN losses on chromosomes 8p and 22q and CN gains on chromosome 9q being the sub-clonal CNAs most frequently observed (Supplementary Data S1).

Bi-allelic CN losses were mainly observed on chromosomes 13q (13/513, 2.53%), 6p and 6q (12/513, 2.34%), 8p and 8q (12/513, 2.34%) and 16q (7/513, 1.36%).

The highest CN gains (up to 6 N) were observed in odd numbered chromosomes, mainly in chromosomes 9, 15 and 19. High CN gains on chromosome 1q were observed in 6.8% (35 samples) of the overall cases (22.2% of patients carrying amp 1q). CN gains in odd-numbered chromosomes preferentially involved chromosomes 19 (46,1%), 9 (45,2%), 15 (43,8%), 11 (38,1%), 5 (37,9%), 3 (36,5%), 7 (28,8%), 21 (22,6%), as ranked by their frequencies. HD, as defined by the presence of CN gains affecting at least two odd numbered chromosomes, was observed in 57.5% of patients (295 samples).

Translocations t-IgH were reported by FISH in 48% of patients, therefore resulting the fourth most frequent genomic lesions in the dataset analysed; in particular, t(11;14)(q23;q32), t(4;14)(p14;q32), t(14;16)(q32;q23), t(14;20)(q32;q12) and t(6;14)(p21;q32) were observed in 23,8%, 22,4%, 5,3%, 1,7% and 0,7% of patients, respectively.

As expected, the whole number of chromosomal aberrations observed in our cohort was the result of three main types of genomic events affecting plasma cells: structural aberrations involving the IgH gene (translocations), scattered numerical aberrations (mainly affecting chromosomes 1 and 13) and aneuploidies (mainly affecting odd-numbered chromosomes). Despite the genomic background of the MM predominant clone emerging at diagnosis was extremely heterogenous, we observed a common genomic ground characterizing all patients, consisting of at least one of the following lesions: HD, loss 13q, t-IgH and gain 1q (Fig. 1c). This observation was confirmed in the CoMMpass dataset, whose chromosomal aberrations' distribution is resumed in Fig. 1d.

### Genomic lesions observed in MM plasma cells tend to co-occur

In most patients, the presence of recurrent genomic aberrations arises within a complex genomic background, with several other co-occurring genomic lesions. The whole catalogue of CNAs was therefore analysed, to identify common pattern of co-occurrence.

A correlation matrix was built, to explore the probability of each genomic aberration (as observed in at least 5% of patients) co-occurring with any other. This highlighted either common or specific associations between variants (Fig. 2a). Since rare genomic lesions causes an excessive scattering of the correlation matrix, they were not included in the figure, nonetheless they were retained in all the subsequent analyses of the present paper.

Overall, areas of significant correlation were observed both among CN gains (average R = 0.42) - particularly in odd-numbered chromosomes (average R = 0.71) - and among CN losses affecting even numbered chromosomes (average R = 0.34); CN gains on chromosomes 15 and 19, 5 and 15, 9 and 15, and CN losses on chromosomes 13 and 14 were the most significantly correlated, among those observed.

Mostly, CN gains and CN losses were significantly inversely correlated; nevertheless, this pattern included a striking exception accounted for CN gains occurring on chromosomes 1q, which tended to correlate with CN losses, the strongest one being with chromosome

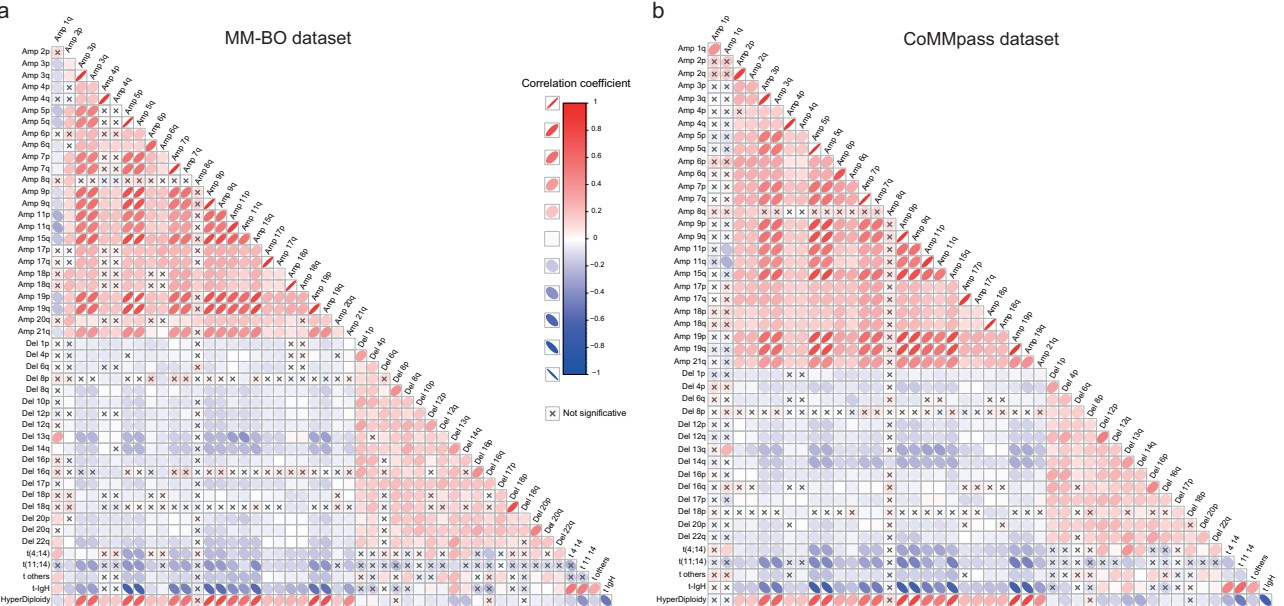

**Fig. 2 | Co-occurrence analysis.** Correlation matrices were built to highlight direct correlations between genomic variables. All aberrations (with frequency>5%) were included in the analysis for (**a**) BO dataset, *n* = 513 patients, and (**b**) CoMMpass dataset, *n* = 752 patients; rare aberrations were excluded from the plot but not from the analyses, to avoid an excessive scattering of the correlation matrix. The heatmap resumed results coming from Pearson correlation tests, performed for each couple of variables, as called when detected within a clonality range between 50% and 100%, as defined above. Direct and indirect correlations were both highlighted in red and blue, and defined by the ovals' orientation (left-oriented and right-oriented ovals highlight direct and indirect correlations, respectively); the ovals' transparency identified the correlation's strength (strong and weak correlations are bold and light coloured, respectively). An X highlighted not-significant correlations (two-sided *p* > 0.05). The approach to multiple testing correction is described in methods section (Co-occurrence analysis).

13q (R = 0.38, *p* < 0.001), followed by those occurring on 20p, 16p, 16q, 18p and 22q, and to inversely correlate with CN gains. In addition, odd-numbered hyperdiploidy appears to be significantly inversely correlated to loss 13q, gain 1q and t-IgH. Finally, both loss 13q and gain 1q correlate to t-IgH. To further support the observed correlations, a Jaccard similarity matrix was generated by using the dichotomized calls of alterations. This analysis is able to capture the intersection between specific pairs of alterations, thus confirming the co-occurrence of gain 1q and loss 13 (Supplementary Fig. S2).

The same results were obtained by performing association matrix on CNAs data extrapolated from CoMMpass dataset (Fig. 2b).

Therefore, even if the genomic landscape of MM patients is quite heterogeneous, patterns of co-occurrence and/or of inverse correlation between chromosomal aberrations might be highlighted, with the most evident being the well-known mutual exclusivity of HD and t-IgH. However, the correlation between gain 1q and CN losses on most chromosomes, among which 13q was the most frequently involved, also appeared highly significant.

**Major component of MM genomic heterogeneity**

The co-occurrence of chromosomal aberrations allowed the development of a matrix-organized view of MM genomic background. However, this approach could not fully describe the complexity of the relationships' network existing between all genomic alterations observed in MM, which is composed by a mix of both co-occourring independent genomic alterations and rare, even if not unique, genomic events.

Therefore, to obtain a broad overview of the relationships between all the genomic variables considered in the previous analysis and to cluster patients according to their own specific complexity (and not just according to the presence of recurrent genomic aberrations) we considered dimensional scaling techniques (i.e., Non-Metric Multi-Dimensional Scaling, NMDS and Principal Component Analysis, PCA) as being appropriate to describe and eventually reduce the complexity of the whole dataset of annotated CNAs. By employing these

approaches, we aimed at discriminating the most informative and, at the same time, most independent genomic events, among those contributing to the MM genomic complexity, and use them to neatly stratify MM patients (Fig. 3).

Input variables for NMDS and PCA included the presence or absence of any CNAs, HD and t-IgH. In order to gather events describing the same level of genomic complexity, CN gains located on odd-numbered chromosomes were merged within the "HD" category while IgH translocations were merged within the "t-IgH" category.

By using NMDS, two mains, partially overlapping clusters of patients were generated, which allocated along crossing axis, as shown in Fig. 3a. As expected, the presence of either HD or t-IgH identified patients belonging to opposite clusters, thus supporting the well-known stratification of HD and non-HD myelomas (these latter, mostly overlapping t-IgH ones). Among the other recurrent CNAs, loss 13q and gain 1q were frequently shown to identify overlapping clusters of patients, located transversely between the other two main clusters, thus supporting the above-mentioned co-occurrence analysis. Strikingly, when considered together, loss 13q and gain 1q identified three well-distinct clusters of patients, named 1q&13+ when both alterations were present, 1q&13- when both were absent and 1q/13 when either one of them was present. These three clusters located perpendicularly with respect to those carrying HD and t-IgH in the low-dimensional space (Fig. 3a, b), thus suggesting an overall independence of this genomic configuration from the conventional HD and t-IgH classification. The same results were obtained by performing NMDS and PCA analyses on data extrapolated from CoMMpass dataset (Fig. 3d, e).

Therefore, the application of MDS techniques to disclose the MM genomic complexity not only recapitulated the well-known stratification of HD and t-IgH patients, but also pinpointed an independent cluster of patients, defined by the presence of both gain 1q and loss 13q, located transversally to the well-known HD−t-IgH (non-HD) axis (Fig. 3c, f). This observation, along with the consistency of the correlation between gain 1q and loss 13q might support the hypothesis that the progression of a discrete sub-group of myelomas, i.e. 1q&13 + ,

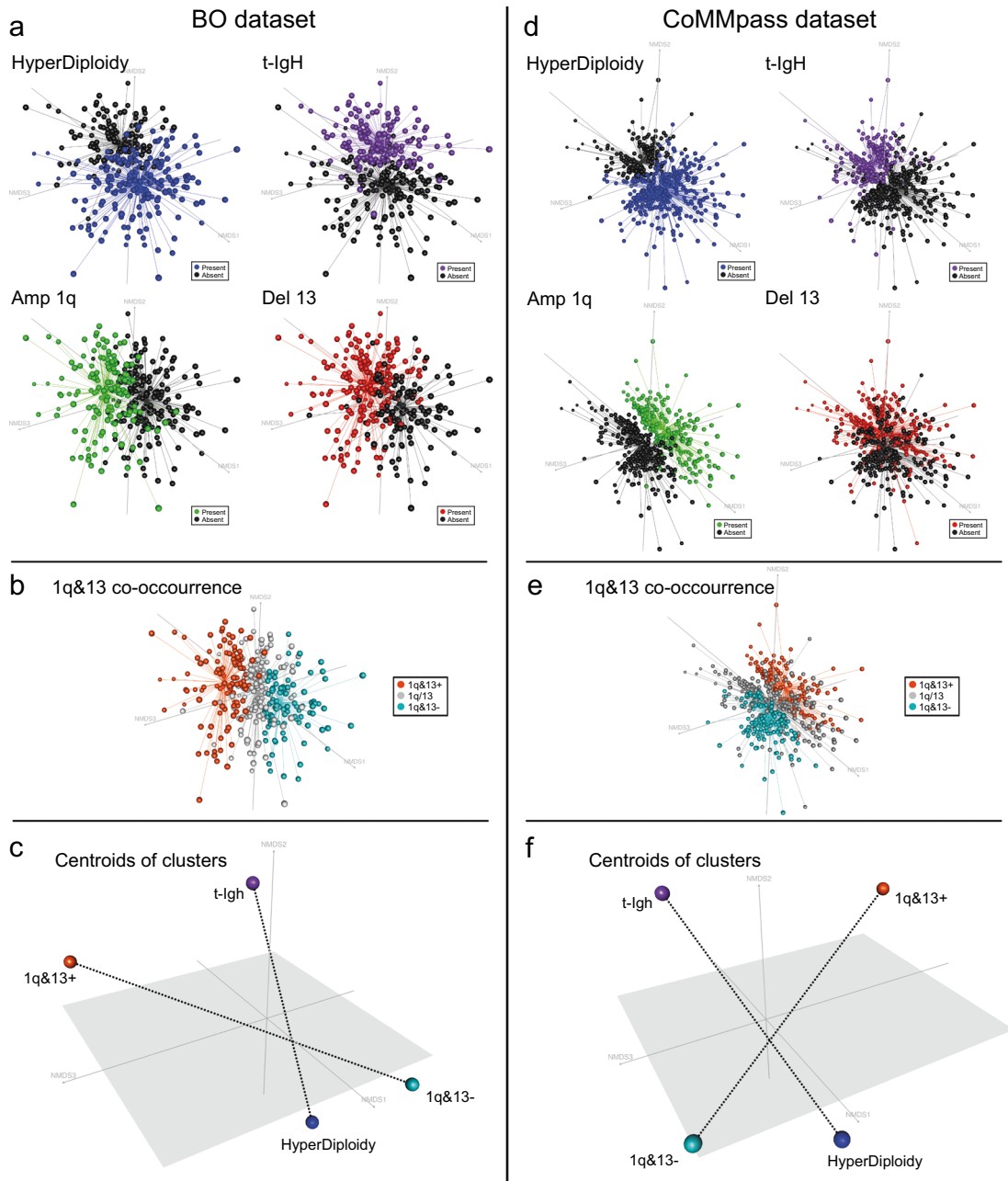

**Fig. 3 | Dimensionality reduction: tridimensional picture of the complex relationships observed between all MM genomic variants.** Each patient was represented by a dot, with cluster of dots —representing patients sharing similar genomic backgrounds—located accordingly in the dimensionality-reduced space. **a, d** Blue and violet clusters, representing patients carrying H and t-IgH, respectively, were located at the opposed extremities of the horizontal axis, whereas red and green clusters (representing patients carrying del13 and amp1q, respectively) tended to overlap and to locate along the vertical axis. **b, e** Orange, grey and teal clusters (representing 1q&13 + , 1q/13 and 1q&13- patients, respectively) identified three well-distinct clusters of patients located transversally with respect to those carrying HD and t-IgH. **c, f** The centroids of 1q&13 + , 1q&13-, HD and t-IgH clusters previously defined are represented, highlighting the transversal positioning of the 1q&13 axis with respect to the HD-IgH axis. Such axes are displayed by dashed lines that connect the centroids.

might occur throughout an independent path, not solely driven by HD nor by t-IgH.

### Gain 1q and loss 13q are driver genomic traits

To further support the hypothesis that the co-occurrence of gain 1q and loss 13q might drive the progression of a discrete sub-group of MM patients, we sought to measure their potential as oncogenic drivers (named "driverness"[18]). Aim of this analysis was to understand whether this peculiar genomic co-occurrence might be considered a "primary" event (i.e., unique to a cell population with the same origin), as for HD

or t-IgH. Accordingly, we defined an event as a "driver" (i.e., conferring a fitness advantage to the tumour population) when it occours frequently in the overall population (penetrance) and it is observed in the context of a "simple" genomic background, thus supporting the tumour progression without the need of for additional complex genomic alterations.

Based on these assumptions, a "Driverness Index" (DI) was developed, which evaluated, in a single score, both the penetrance of a given genomic alteration ($\rho$) and the median genomic complexity of all samples carrying this alteration (MTGC or Median Total Genomic

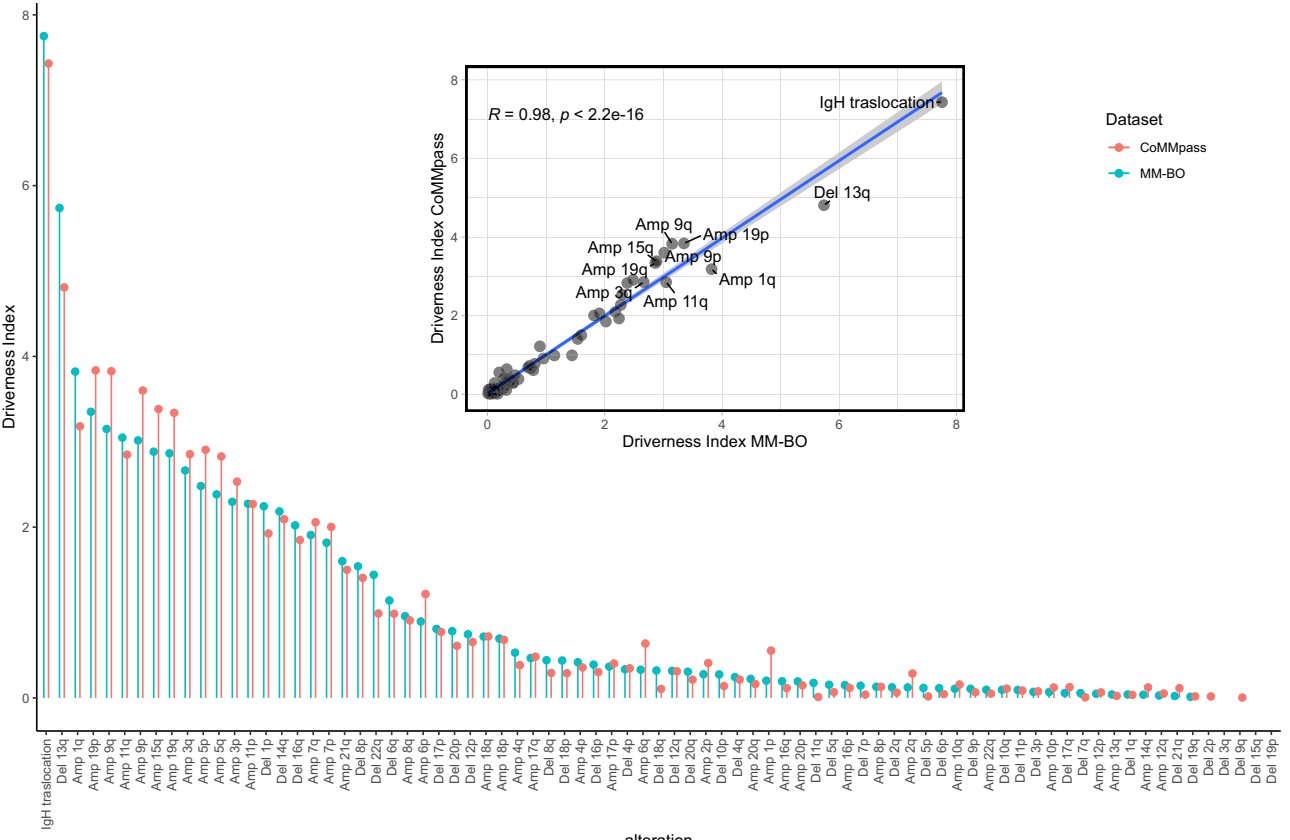

**Fig. 4 | Lollipop plot representing the Driverness Index (DI) score for each alteration, as evaluated both in the BO- (blue dots, *n* = 513 patients) and in the CoMMpass datasets (orange dots, *n* = 752 patients).** Alterations were sorted along the x axis accordingly to the BO-dataset's DI. In the nested plot the CoMMpass DI scores are shown to be highly similar to BO DI scores (Spearman's rho= 0.942, two-sided *p* = 8.73e−36), as illustrated by the closeness of the data points to the blue linear regression line. The 95% confidence interval error band of the linear regression line is represented by a shaded grey area around the line.

Complexity) (Eq. 1).

$$DI = \frac{\rho}{MTGC} \qquad (1)$$

Accordingly, a driver alteration was expected to show a high ρ and a low MTCG, while a passenger alteration was expected to show a low ρ and a high MTGC. Therefore, the higher the DI resulted, the more the genetic alteration was considered to be an oncogenic driver. DI was evaluated in both BO and CoMMpass datasets and results are summarized in Fig. 4. Trisomies in odd-numbered chromosomes, t-IgH, loss 13q and gain 1q resulted the top-driver aberrations, well separated from the bulk of other alterations, presenting a much lower DI.

Therefore, the occurrence of both gain 1q and loss 13q, matching HD and t-IgH, seemed to be driver events in the progression of MM, affecting a discrete sub-group of patients.

### Genomic context of patients carrying chromosome gain 1q and loss 13q: 1q&13 classification

The occurrence of both gain 1q and loss 13q affected 131/513 (26%) patients included in the study (collectively named "1q&13 +"); on the contrary, 212/513 (41%) patients did not carry any of these two CNAs (collectively named "1q&13-"); the remaining patients carried either gain 1q only, 46/513 (9%), or loss13q only, 124/513 (24%) (collectively named "1q/13"). Hereinafter we refer to these groups with the overall term "1q&13 classification", Fig. 5a−c.

The overall genomic complexity, as defined by the number of chromosomal events *per* patient, was higher in 1q&13+ patients, as compared to both 1q&13- and 1q/13 subgroups (MTGC: 7.07 (CI 95% = 6.51−7.63), 5.20 (CI 95% = 4.73−5.66) and 3.19 (CI 95% = 2.85−3.54) in 1q&13 +, 1q&13- and 1q/13, respectively, Kruskal-Wallis *p* < 0.001) (Fig. 5d). In detail, HD was over-represented in 1q&13- patients, whereas 1q&13+ patients were characterized by the presence of 1p CN losses and 17p CN losses genomic events. Translocations t(4;14)(p16;q32) and t(14;16)(q32;q23) were more prevalent in 1q&13+ patients, whereas t(11,14)(q13;q32) was under-represented among these patients. In addition, along with gain 1q and loss 13q, several focal aberrations enriched in 1q&13+ patients, e.g., CN losses on chromosome 14q (covering *TRAF3*) and on chromosome 16q (covering *CYLD*) (Supplementary Data S2). A similar distribution of CNAs and t-IgH was confirmed among patients included in the CoMMpass study (Supplementary Data S3), stratified according to the *1q&13* classification, thus supporting the overall reported chromosomal aberrations' distribution.

### Transcriptomic context of 1q&13+ patients

To further characterize the biological context of patients classified as described above, we also analysed their expression profiles. To this aim we employed RNA-seq data collected in the context of the CoMMpass study.

Raw expression data were first analyzed using MDS techniques, revealing two main transcriptomes clusters driven either by the presence or the absence of both gain 1q and loss 13q (Fig. 6a).

We then compared the expression profiles of all the combinations of the three 1q&13 classes, showing that the only comparison resulting in a list of significantly deregulated genes was 1q&13+ *versus* 1q&13-

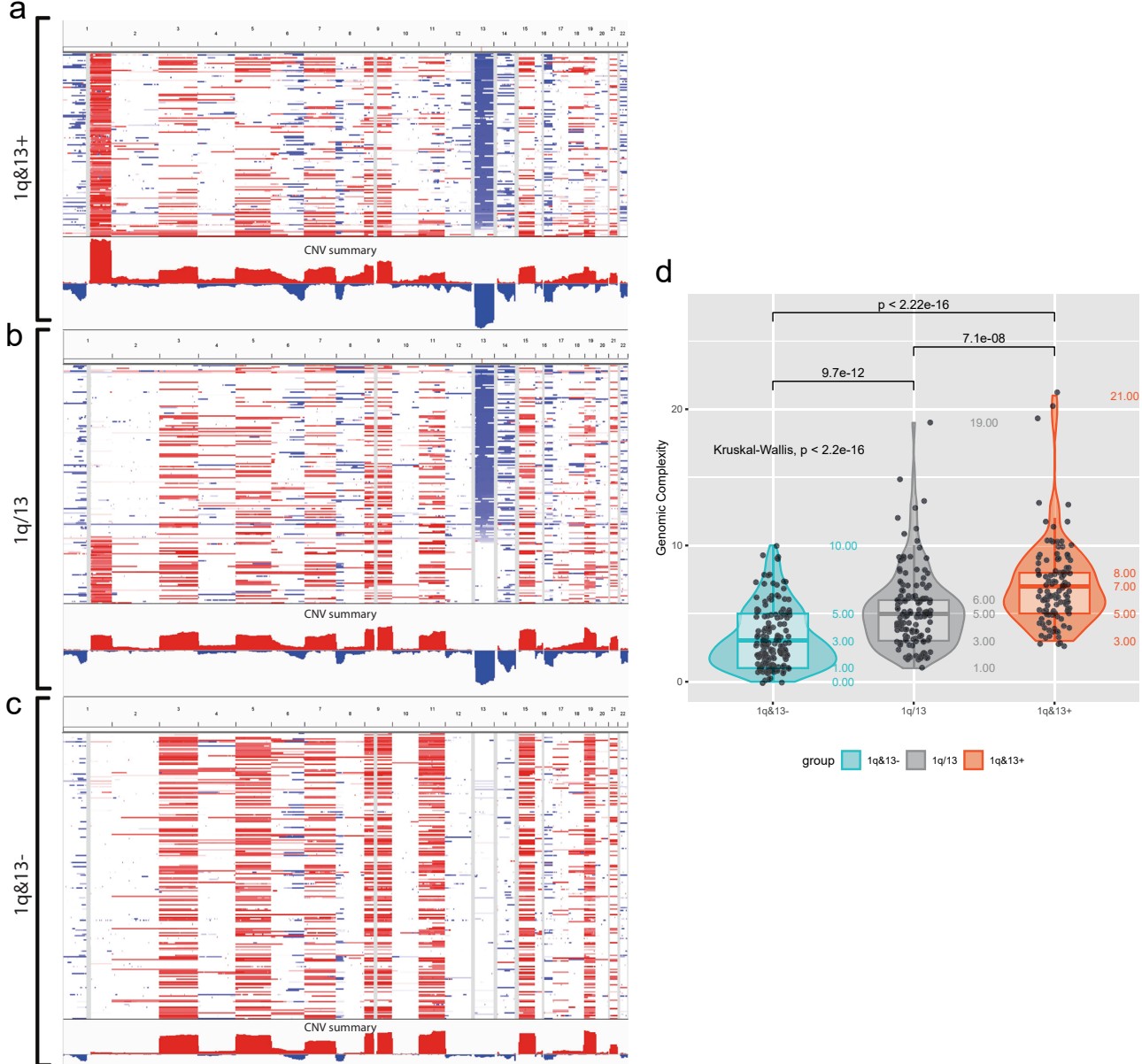

**Fig. 5 | Genomic context of patients stratified according to the "1q&13 classification". a–c** Showed the CNAs profile of 1q&13 + , 1q/13 and 1q&13- patients, as obtained by IGV screenshots; CN gains and CN losses were highlighted in red and blue, respectively. **d** Box-Violin plot showing the statistically different genomic complexity distribution among the three subgroups of patients (Kruskal-Wallis test, *p* = 1.39e−19). Two-sided Mann–Whitney test was used to test subgroups pairwise differences. The box plot within each violin shows the median (central line), first quartile (bottom edge of the box), and third quartile (top edge of the box). The numbers within or near each plot indicate the minimum, first quartile, median, third quartile, and maximum number of chromosomal aberrations per subgroup. In this analysis, *N* = 512 independent patients were examined, each in a single independent experiment.

(Supplementary Data S4). The volcano plot drawn in Fig. 6b showed that 301 genes were significantly differentially expressed among the two patients' groups (q value < 0.05, Fold Change, FC ≥ ± 2). Of note, the most significantly up and down regulated genes in 1q&13+ patients were *CCND2* and *CCND1*, respectively, a pattern confirmed also by excluding from the analysis all patients carrying t-IgH, which are known to cause the cyclin D overexpression in MM[19] (Supplementary Fig. S3a).

To get insight into the expression profiles of 1q&13+ patients, we first confirmed the mutually exclusive expression of *CCND1, CCND2* and *CCND3*, as already frequently reported[19–21], and annotated patients accordingly (Supplementary Fig. S3b). We then performed a non-supervised clustering analysis, according to the expression of 7 genes, selected as being significantly de-regulated in patients carrying t-IgH (i.e., *NSD2, FGFR3, CCND1, CCND2, CCND3, MAF* and *MAFB*)[22] Fig. 6c).

Notably, most 1q&13+ patients clustered among those who up-regulated *CCND2* (*p* < 0.001), whereas 1q&13- patients clustered among those who up-regulated *CCDN1* (*p* < 0.001) (Fig. 6c). Moreover, 58,9% of *1q&13+* patients (89/151) up-regulated *CCND2*, even though they did not carry t(4;14), nor t(14;16) or t(14;20).

On the contrary, only 19,6% of 1q&13- patients (52/265) up-regulated *CCND1*, even though they did not carry t(11;14)(q23;q32) nor chromosome 11q trisomy (Fig. 6c).

Therefore, the mutually exclusive expression of *CCND1* and *CCND2* seemed to be mainly driven by the 1q&13 classification.

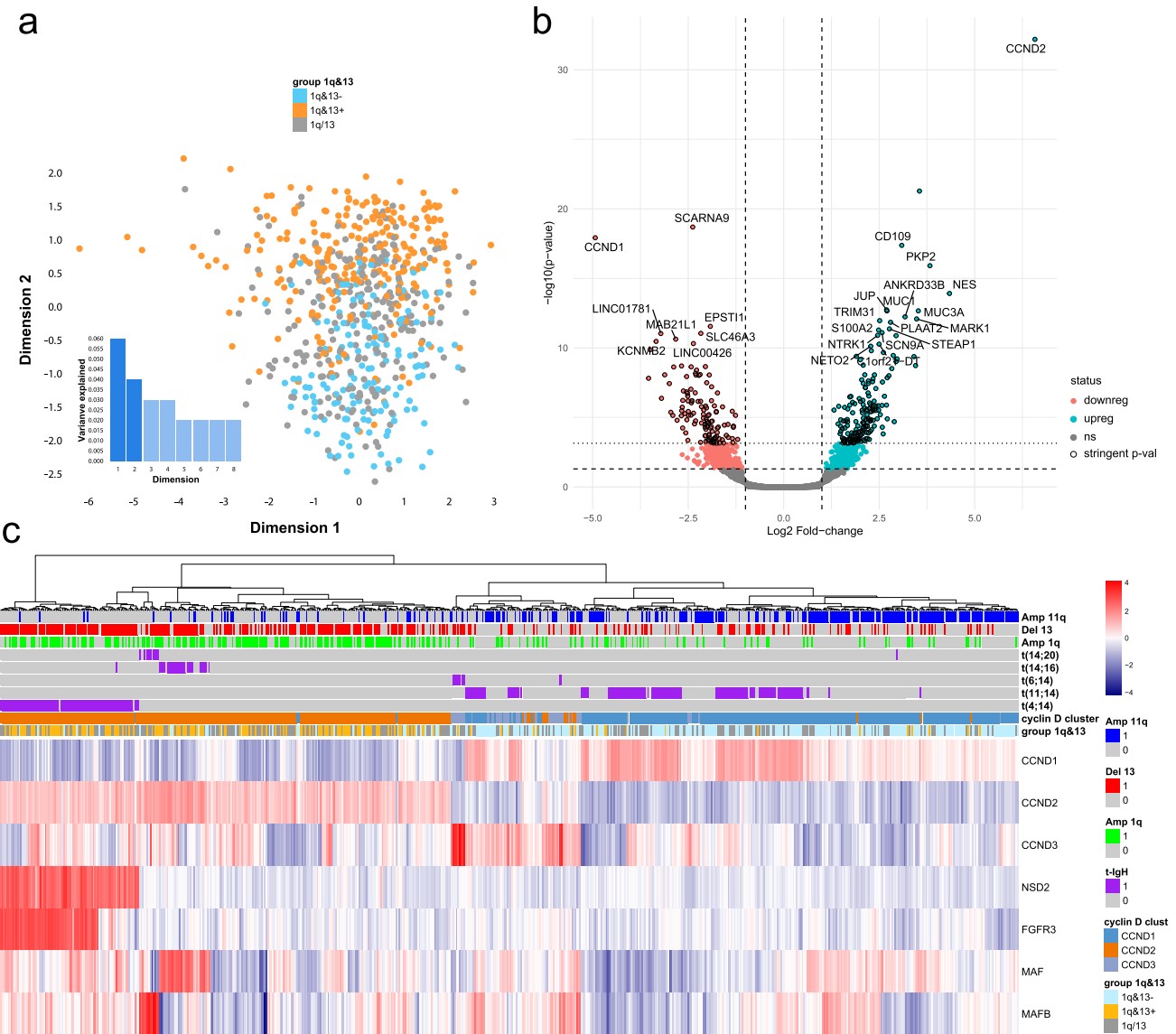

**Fig. 6 | Gene expression profile analysis (CoMMpass dataset).** RNA-seq, gene-based data, downloaded from CoMMpass IA14 were used to validate the biological significance of 1q&13 classification (*n* = 659 independent samples). **a** Raw transcriptome data dimensionality reduction analysis. Every dot represents a different sample: red= 1q&13 + , green=1q&13-, blue=1q/13; separate and opposite clusters were highlighted also at expression level. **b** Volcano plot of the 301 genes differentially expressed between 1q&13+ and 1q&13-. Stringent-significative and significative genes, as defined by Bayes-moderated *t* tests ("treat" and "eBayes" limma functions, respectively—additional details in "Methods") are highlighted in cyan if upregulated and in pink if downregulated. The 301 stringent-significative genes are highlighted by a black stroke around the points. **c** Heatmap describing the unsupervised clustering of 7 genes representative of t-IgH expression profile. Columns represent patients. Colours are scaled row-wise according to the normalized expression values of genes. The 1q&13 classification is plotted over the heatmap (orange = 1q&13 + , blue= 1q&13-, green = 1q/13 + ).

Finally, to identify gene sets significantly over-represented in 1q&13 + , as compared to 1q&13- patients, the list of genes differentially expressed between the two sub-groups of patients (1795 genes with FC ≥ ± 1) was used to perform a Gene Set Enrichment Analysis (GSEA). Overall, in 1q&13+ patients 32 and 96 gene sets were significantly enriched for positive and negative Enrichment Score (ER) respectively, with a False Discovery Rate (FDR) < 25% (Supplementary Data S5).

Two main groups of genes emerged as the most significantly upregulated in 1q&13+ as compared to 1q&13- patients: the first one included, among the others, *CD109* (FC = 3,1) and *YAP1* (FC = 1,6), both involved in the epithelial-mesenchymal transition process[23], *CTSK* (FC = 1,1), involved in osteoclast formation process and over-expressed in several cancers[24], and *JUP* (FC = 2,7), a cell-cell junction protein,

homologue of β-catenin, involved in adhesion junction and desmosome composition[25]; the second one included, among the others, *CCND2* (FC = 6,6), *NES* (FC = 4,3), *NEK2* (FC = 1,7), *CDKN2B* (FC = 1,2) and *CDK6* (FC = 1,2), all involved in cell cycle process. Genes downregulated in 1q&13+ as compared to 1q&13- patients mainly pertained to chemotaxis and cell adhesion pathways (e.g., cell chemotaxis and regulation of cell adhesion), and to the regulation of inflammatory response. Among the others, several chemokines family's receptors and ligands (e.g., *CCRL2, CXCR2, CXCL16, CXCL1, CXCL12, CCL8, CCL25, CCL2, CCL13, CCR5, CCR1, CCR6*) were significantly downregulated (FCs ranging from -1.1 to -2,3), as well as genes involved in angiogenesis pathway (e.g., *NRP1, TIAM1*, FC = -1,4 and -1,5, respectively) and inflammatory response (e.g., *STAP1, MDK*, FC = -1,9 and -1,6, respectively) (Supplementary Data S6).

## Clinical outcome of 1q&13+ patients

We finally analyzed the clinical behaviour of 1q&13+ patients. The overall cohort of MM included three subgroups of patients with clinical data (named "BO-1", "BO-2" and "BO-3") with different median follow-ups and treated with different regimens, as described in Table 1. Patients were either previously enroled in the EMN02 or in BO2005 clinical trials, whose results has been previously reported[26,27], or consecutively treated in our Institution as part of routine clinical practice.

The baseline clinical characteristics of the three subgroups of patients are described in the Patients and Methods section, as well as details concerning their therapeutic treatment and their overall clinical outcomes. Despite the differences between the three subgroups of patients, we chose to merge all patients in the final clinical analysis, yet by stratifying on subgroups' origin, both to increment the statistical analysis' power and to control any possible bias derived from the unbalanced distribution of clinical features. The same statistical approach was employed to analyze the clinical outcomes of patients included in the CoMMpass trial.

As shown in Table 2, the 1q&13+ patients' baseline clinical characteristics highlighted a biased over-representation of clinical and genomic high-risk features (e.g., ISS stage 3, Albumin<3.5 g/dL, FISH del 17p, FISH del 1p), as compared to those of the rest of patients. Similarly, patients included in the CoMMpass trial, stratified according to 1q&13 classification, showed an unbalanced distribution of baseline clinical characteristics (Supplementary Data S7). In addition, parameters associated to higher proliferation rate and to disease invasiveness, such as Circulating MM Cells (CMMCs) count, evaluated for most patients enroled in CoMMpass trial, proved significantly higher in 1q&13+ patients, as compared to the others (median CMMCs count: 4147 vs. 1292 vs. 708, $p < 0.0001$) (Supplementary Data S7). Therefore, 1q&13+ patients displayed well-known baseline high-risk features, commonly correlated with bad prognosis, thus suggesting that this genomic-based patients' stratification can describe not only a biological but also a clinical specific condition.

To confirm this hypothesis, Kaplan-Meier survival analyses were performed, and Cox-hazard models were built on patients stratified according to the 1q&13 classification (model 1). Both datasets (BO and CoMMpass) were independently analyzed, in order to validate the results, showing that overall 1q&13+ patients' progression-free (PFS) and overall survivals (OS) were significantly shortened, as compared to those of the other subgroups of patients (BO dataset 5-year PFS: 22% vs. 37% vs. 47%, $p < 0.0001$; OS: 50% vs. 74% vs. 78%, $p < 0.001$ for 1q&13 +, 1q/13, 1q&13- respectively. CoMMpass dataset 3-year PFS: 37% vs. 43% vs. 55%, $p = 0.0003$; OS: 67% vs. 76% vs. 81%, $p = 0.002$ for 1q&13 +, 1q/13, 1q&13- respectively) (Fig. 7a–d).

In addition, PFS and OS were significantly shortened in 1q&13+ patients as well as compared to those of patients carrying either gain 1q or loss 13q alone, both in BO and CoMMpass datasets (Supplementary Fig. S4).

In a multivariate Cox model, the risk conferred by the co-occurrence of gain 1q and loss 13q was both similar to and independent from those conferred either by t(4;14)(p16;q32) or by del(17p), as well as independent from any other considered risk factors (e.g. being ISS3), both in terms of PFS and OS; similar results were obtained in the CoMMpass dataset (Fig. 8a–d). In particular, we observed that 1q&13+ patients were distributed across all ISS (and R-ISS for CoMMpass) classes and were even present in the low risk ISS1 (and R-ISS I) class, raising the question whether a possible inclusion of 1q&13 classifier into the current stratification systems might be considered an added value to improve the high-risk definition in MM (Supplementary Table S1).

Finally, since 47,3% (62/131) of 1q&13+ patients carried also *CCND2* or *MAF* deregulating t-IgHs (i.e., t(4;14), t(14;16) and t(14;20), Fig. 6c), we evaluated in novel Cox multivariate models (model 2), the risk of relapse and death conferred by this genomic combination (named "t&1q&13 +"). We showed that, even though quite rare - observed in 62/513 (12,1%) patients in BO dataset and in 75/840 (8,9%) in CoMMpass dataset - this genomic combination impacted patients' outcome independently and more heavily with respect to any other genomic lesion, both in terms of PFS and OS (Fig. 8e–h). Similar results were obtained in the CoMMpass dataset, where we compared the Hazard Ratios (HR) of t&1q&13+ and R-ISS3 patients (Supplementary Fig. S5). Notably, to assess R-ISS' HR, we built a new Cox hazard model, since t(4;14) and t(14;16) are both included in R-ISS and in t&1q&13+ stratifications. Thus, it was not possible to directly compare the two groups in the same multivariate model, to avoid statistical collinearity effect.

In this clinical context, we showed that the risk conferred by t&1q&13+ was higher, as compared to that conferred by R-ISS3, both in PFS and OS: PFS 1,56 (CI 1.02–2.37) vs. 1,40 (CI 0,98-1,99), OS 2,04 (CI 1.14–3.67) vs. 1,74 (CI 1.09–2.76) in t&1q&13+ vs. ISS-3 HR patients, respectively.

## Discussion

In recent years, the limitation of the dichotomic stratification of MM patients in two main subgroups (HD and non-HD) has become increasingly clear, thanks to the use of high-throughput, genome-wide molecular approaches, showing that, on the contrary, several well-defined genomic subgroups might compose the MM patients' population[2]. Nevertheless, the MM complexity still remains not fully described by most of the patients' stratifications proposed so far, which overall describe the complex interactions' network between chromosomal aberrations throughout a limited and supervised one-to-one superstructure.

To overcome this issue, in this paper we explored the genomic heterogeneity of MM with dimensionality-reduction techniques, which take into consideration all the possible interactions between every detected chromosomal alteration, thus thoroughly describing the overall complexity of MM patients. Moreover, this bioinformatic approach highlighted the genomic variables which can most significantly describe the overall MM complexity. It also enabled the exploration of the unsupervised distribution of patients in the low-dimensionality space, driven just by their genomic similarities. According to the innovative bioinformatic approach employed, the global structure of interactions existing between aberrations has been deeply explored, highlighting that, not only patients carrying either HD or non-HD alterations cluster together, by forming two opposite subgroups, but also that a third cluster of patients emerged, carrying both gain 1q and loss 13q, hence named 1q&13 +. This latter cluster has been so far hidden within the conventional dichotomic distribution of chromosomal aberrations, since it is located in a specific and distinct position in the space, across the plane defined by HD and non-HD clusters, partially overlapping these clusters, but clearly distinguishable from them. Notably, this patients' distribution was confirmed also on CNAs data derived from the CoMMpass dataset.

The use of orthogonal unsupervised methodologies to determine whether MM genomic profiles could specify meaningful genetic and clinical subgroups has already been explored, identifying, for instance, one distinct genomic pattern characterized by the co-occurrence of HD, gain 1q and loss 13q among those described in a cohort of 67 MM cases analyzed by aCGH (array comparative genomic hybridization)[28]. Of note, this subgroup of patients was characterized by the deregulated expression of pathways consistent with a more advanced progression of MM, as also confirmed by the bad prognosis of this subgroup of patients. Similarly, the co-occurrence of gain 1q and loss 13q has been also more recently observed[8,29,30], even though the genomic and clinical features associated to this particular chromosomal configuration have not been explored.

The identification of primary cytogenetic abnormalities, unique to a cell population with the same origin, might favours the accurate

**Table 1 | All the clinical and baseline variables available to describe the "BO dataset" were included in this study**

| Variable | BO-1 (n) | n/N | Median | IQR | BO-2 (n) | n/N | Median | IQR | BO-3 (n) | n/N | Median | IQR | P value |
|---|---|---|---|---|---|---|---|---|---|---|---|---|---|
| Male | 214 | 58% | – | – | 41 | 56% | – | – | 47 | 67% | – | – | ns |
| Female | 156 | 42% | – | – | 32 | 44% | – | – | 23 | 33% | – | – | ns |
| Age, years | 370 | 100% | 58 | 52–62 | 72 | 99% | 66 | 58.75–72.25 | 70 | 100% | 59 | 54–62 | <0.0001 |
| Age >65 years | 4 | 1% | – | – | 38 | 53% | – | – | 0 | 0% | – | – | <0.0001 |
| Beta 2 microglobulin, mg/L | 370 | 100% | 3.5 | 2.433–5.2 | 65 | 89% | 4.3 | 3.2–6.8 | 70 | 100% | 3.17 | 2.4–4.6 | 0.004 |
| Beta 2 microglobulin <3.5 mg/L | 183 | 49% | – | – | 25 | 38% | – | – | 47 | 67% | – | – | 0.003 |
| Beta 2 microglobulin >5.5 mg/L | 85 | 23% | – | – | 23 | 35% | – | – | 9 | 13% | – | – | 0.008 |
| Albumine, g/dL | 370 | 100% | 3.8 | 3.39–4.2 | 64 | 88% | 3.8 | 3.375–4.1 | 70 | 100% | 3.94 | 3.51–4.28 | ns |
| Albumine <3.5 g/dL | 110 | 30% | – | – | 21 | 33% | – | – | 16 | 23% | – | – | ns |
| Creatinine, mg/dL | 367 | 99% | 0.9 | 0.71–1.1 | 72 | 99% | 0.99 | 0.78–1.337 | 70 | 100% | 1 | 0.8–1.2 | 0.009 |
| Haemoglobin, g/dL | 370 | 100% | 10.95 | 9.6–12.4 | 72 | 99% | 10.5 | 9.4–11.7 | 70 | 100% | 11.3 | 9.9–12.2 | ns |
| Haemoglobin <10.5 g/dL | 155 | 42% | – | – | 36 | 50% | – | – | 27 | 39% | – | – | ns |
| Platelet count, 10^3/mL | 370 | 100% | 230.5 | 176–278 | 71 | 97% | 214 | 176.5–275.5 | 70 | 100% | 225 | 183.2–285.8 | ns |
| Platelet count <150 10^3/mL | 51 | 14% | – | – | 11 | 15% | – | – | 7 | 10% | – | – | ns |
| Lactate dehydrogenase, g/dL | 355 | 96% | 450 | 240.5–480 | 56 | 77% | 166.5 | 142–207.25 | 65 | 93% | 265 | 205–349 | <0.0001 |
| LDH, Upper Limit | 101 | 39% | – | – | 7 | 15% | – | – | 47 | 82% | – | – | <0.0001 |
| Bone marrow plasma cells >60% | 205 | 57% | – | – | 26 | 45% | – | – | 37 | 59% | – | – | ns |
| IG Isotype IgG | 219 | 62% | – | – | 42 | 60% | – | – | 45 | 64% | – | – | ns |
| IG Isotype IgA | 73 | 21% | – | – | 14 | 20% | – | – | 13 | 19% | – | – | ns |
| IG Isotype BJ | 53 | 15% | – | – | 13 | 19% | – | – | 12 | 17% | – | – | ns |
| Light Chain Kappa | NaN | – | – | – | 44 | 65% | – | – | 37 | 60% | – | – | ns |
| Light Chain Lambda | NaN | – | – | – | 24 | 35% | – | – | 25 | 40% | – | – | ns |
| ISS I | 139 | 38% | – | – | 22 | 33% | – | – | 31 | 45% | – | – | ns |
| ISS II | 143 | 39% | – | – | 20 | 30% | – | – | 28 | 41% | – | – | ns |
| ISS III | 88 | 23% | – | – | 25 | 37% | – | – | 10 | 14% | – | – | 0.008 |
| R-ISS I | 56 | 16% | – | – | 8 | 18% | – | – | NaN | – | – | – | ns |
| R-ISS II | 241 | 69% | – | – | 28 | 64% | – | – | NaN | – | – | – | ns |
| R-ISS III | 51 | 15% | – | – | 8 | 18% | – | – | NaN | – | – | – | ns |
| t(4,14) | 58 | 16% | – | – | 6 | 9% | – | – | 26 | 38% | – | – | <0.0001 |
| t(6,14) | 3 | 1% | – | – | 0 | 0% | – | – | NaN | – | – | – | ns |
| t(11,14) | 80 | 23% | – | – | 1 | 3% | – | – | NaN | – | – | – | 0.0004 |
| t(14,16) | 18 | 5% | – | – | 4 | 7% | – | – | NaN | – | – | – | ns |
| t(14,20) | 7 | 2% | – | – | 0 | 0% | – | – | NaN | – | – | – | ns |
| FISH Deletion 13 | 182 | 52% | – | – | 14 | 28% | – | – | 26 | 46% | – | – | 0.007 |
| FISH Deletion 17p | 41 | 12% | – | – | 12 | 18% | – | – | 3 | 5% | – | – | ns |
| FISH Deletion 1p36 | 47 | 13% | – | – | 7 | 12% | – | – | NaN | – | – | – | ns |
| FISH Amplification 1q | 137 | 39% | – | – | 21 | 34% | – | – | 10 | 42% | – | – | ns |
| FISH Hyperdiploidy | 170 | 49% | – | – | 8 | 17% | – | – | NaN | – | – | – | <0.0001 |
| Induction (PI) | 370 | 100% | – | – | 22 | 31% | – | – | NaN | – | – | – | <0.0001 |
| Induction (Imid) | NaN | – | – | – | 9 | 13% | – | – | 25 | 36% | – | – | <0.0001 |
| Induction (PI - Imid) | NaN | – | – | – | 37 | 51% | – | – | 45 | 64% | – | – | <0.0001 |
| Induction (Other) | NaN | – | – | – | 4 | 6% | – | – | NaN | – | – | – | <0.0001 |
| ASCT | 211 | 62% | – | – | 39 | 61% | – | – | 64 | 94% | – | – | <0.0001 |
| Single ASCT | 108 | 32% | – | – | 20 | 31% | – | – | 10 | 15% | – | – | 0.01 |
| Double ASCT | 103 | 30% | – | – | 19 | 30% | – | – | 54 | 79% | – | – | <0.0001 |
| Maintenance | 271 | 84% | – | – | 30 | 51% | – | – | 47 | 70% | – | – | <0.0001 |
| Consolidation | 118 | 37% | – | – | 29 | 49% | – | – | 54 | 81% | – | – | <0.0001 |
| Median Follow-up | 370 | 100% | 72 | 62–83 | 72 | 99% | 38 | 26–100 | 70 | 100% | 124 | 119–132 | <0.0001 |

**Table 1 (continued) | All the clinical and baseline variables available to describe the "BO dataset" were included in this study**

| Variable | BO-1 (n) | n/N | Median | IQR | BO-2 (n) | n/N | Median | IQR | BO-3 (n) | n/N | Median | IQR | *P* value |
|---|---|---|---|---|---|---|---|---|---|---|---|---|---|
| Progression Free Survival | 370 | 100% | 39.5 | 16–68 | 72 | 99% | 22.5 | 13–43 | 70 | 100% | 48.5 | 23.2–83.8 | <0.0001 |
| Overall Survival | 370 | 100% | 64 | 34–78 | 72 | 99% | 27.5 | 11–31.5 | 70 | 100% | 80.5 | 551.2–128.8 | <0.0001 |

The number and percentage of patients' data available for any given variable, along with the median value and inter-quantile range (IQR) are showed here, broken down for each of the three cohorts that composes the dataset ("BO-1", "BO-2" and "BO-3"). For categorical variables, two-sided Fisher's Exact test *p* values were used, and for continuous variables, two-sided Wilcoxon-Mann-Whitney test *p* values were computed and shown in the last column.

classification of tumours, by identifying subgroups sharing the same temporal sequence of genetic aberrations, and therefore possibly explaining the process of their specific disease progression. For a long time, the acquisition of either HD or t-IgH has been considered a mutually exclusive, primary event in MM pathogenesis[8]. However, more recently, also gain 1q was shown to be generally acquired together with other trisomies in the earliest time-window during myeloma progression[31].

Here, we suggest that the co-occurrence of gain 1q and loss 13q might represent a third primary event in MM, since these genomic aberrations frequently arise at the asymptomatic phase of the disease[32,33] and are mostly clonal at diagnosis (Fig. 3). To support this concept, we evaluated the "driverness" of the different genomic configurations observed in our dataset, measuring a Driverness Index. This analysis showed that, along with t-IgH and odd-numbered trisomies, both gain 1q and loss 13q were the top earliest chromosomal aberrations.

Overall, in 25.5% of NDMM patients, gain 1q and loss 13q co-occurred and these patients were characterized both by a significantly higher number of aneuploidies affecting several chromosomes, and by the recurrence of t(4;14)(p16;q32). Notably, the association between both gain 1q and loss 13q with t(4;14) has been commonly reported[10,34–36]. However, the recurrence of t(4;14) among 1q&13+ patients (34%), although higher than that observed among 1q&13- (11%) and 1q/13 (15%) patients, yet suggests that the translocation t(4;14) might not be the driver event in 1q&13+ patients, even though some of them might acquire it early during the disease progression.

A further insight into the biological significance of this patients' stratification was provided by the transcriptome profiles analysis of 1q&13+ as compared to 1q&13- patients, as extrapolated by the CoMMpass RNAseq dataset: strikingly, the most significantly over-expressed and down-regulated genes in 1q&13+ patients were *CCND2* and *CCND1*, respectively. The deregulated expression of cyclins D in MM has been demonstrated since many years[17], as well as the fact that the different cyclins genes are mutually exclusively expressed in each MM patient[37]: in fact, patients expressing *CCND1* do not express *CCND2* and vice versa. In addition, a mutually exclusive pairing of cdk4 with cyclin D1 and cdk4/6 with cyclin D2 has been observed to lead to Rb phosphorylation, which is critical in controlling the cell cycle dysregulation in MM[17].

One of the most recognized MM patients stratification, based on both cyclin D expression and the presence of either one of t-IgH translocation (TC classification[37]), has also shown that *CCDN2* is mainly over-expressed in patients carrying t(4;14), whereas *CCND1* in those carrying t(11;14). This has led to a causative association between t(4;14) and *CCND2* overexpression, which, however, has not been demonstrated. Here, we showed that the dichotomic expression of *CCND2* and *CCND1* is almost superimposable to either the presence or the absence of both 1q gain and chromosome 13 loss, and, among 1q&13+ patients, we observed the over-expression of *CCND2*, not only caused by the presence of t(4;14) (Fig. 7). Therefore, we showed that, the grouping of MM patients according to 1q&13 classification might recapitulate the results of transcriptome-based patients' stratification: in fact, 1q&13+ subgroup of patients corresponds to those classified as proliferation-associated genes (PR), translocation cluster (MF) and

bone disease (LB) by the most recent TC classification, whereas 1q&13- subgroup, overexpressing *CCND1*, corresponds to hyperdiploid (HY), CD-1 and CD-2 subgroups of patients, defined low risk and at good prognosis[22]. We finally tried to identify other key genes and pathways that may contribute to describe the biological features of 1q&13+ patients, by performing a pathway enrichment GSEA analysis. Overall, results suggest that 1q&13+ plasma cells express a highly proliferative phenotype, with altered relationship with the bone marrow niche, supporting the possible spread of the disease from the local microenvironment boundaries, as observed in CoMMpass dataset.

To explore the clinical impact of the proposed patients' stratification, we built two prognostic models, the first one exploring the clinical outcome of patients carrying gain 1q and loss 13q, the second one focusing on a smaller subgroup of patients, carrying also *CCND2*&*MAF*-deregulating t-IgHs. Both models were able to identify homogenous subgroups of MM patients, whose outcome was worse than that of the others, as shown by the survival analyses performed both in the BO and in the CoMMpass datasets. Notably, the prognostic impact of both genomic combinations resulted independent from any other patients' baseline clinical characteristics, including the conventionally defined high-risk cytogenetic features. An important difference among the two models resides in the number of patients defined at high-risk, higher in model 1 than in model 2 (approximately 25% and 10% of the whole population, respectively), that indeed represent a subgroup of patients included in model 1.

Overall, the proposed classifications show the importance of studying the genomic landscape of plasma cells to understand different subgroups of patients with varying prognoses. According to the most commonly employed risk scoring systems[14,38–41], about 20% of MM patients are considered high risk for progression, but these systems don't always classify the same patients as high risk[12,42]. This could cause misclassifications and hinder risk-based treatment approaches. Scoring systems often rely on statistical approaches and a few standalone chromosomal abnormalities detected by FISH, without considering their interaction. For example, del(17p) is usually considered high risk, but it may not always affect the time to first progression, especially if it is monoallelic or sub-clonal[43]. The negative impact of del(17p) on prognosis may also depend on the genomic context and the presence of other genomic abnormalities. Similarly, the prognostic role of gain 1q is not fully understood without considering the genomic background, which may include other abnormalities like del(13) and t(4;14)(p16;q32), whose co-occurrence can lead to the de-regulation of specific pathways, causing the expression of a very aggressive plasma cells phenotype.

In addition, the identification of 1q&13 distinct segments of MM, whose underlying biology might explain patients' prognosis, might unveil pathology-related vulnerabilities, possibly therapeutically targetable. The value of this biologically-based patients' stratification will be further deepened by formally comparing the two models proposed here to the conventionally employed scoring systems, in order to support the identification of highly precise high-risk biomarkers.

In conclusion, we showed that the implementation of dimensionality-reduction techniques for the analysis of CNAs genomic profile of a large cohort of newly diagnosed MM patients allowed to define a segment of patients carrying a homogeneous genomic and

**Table 2 | All the clinical and baseline variables available for describing the "BO dataset" were included in this study**

| variable | 1q&13+ (n) | n/N | Median | IQR | 1q/13 (n) | n/N | Median | IQR | 1q&13- (n) | n/N | Median | IQR | P value |
|---|---|---|---|---|---|---|---|---|---|---|---|---|---|
| Male | 63 | 48% | - | - | 95 | 56% | - | - | 144 | 68% | - | - | 0.0008 |
| Female | 68 | 52% | - | - | 75 | 44% | - | - | 68 | 32% | - | - | 0.0008 |
| Age, years | 130 | 99% | 57 | 51-63 | 170 | 100% | 58 | 52-63 | 212 | 100% | 59 | 54-63 | ns |
| Age >65 years | 11 | 8% | - | - | 10 | 6% | - | - | 21 | 10% | - | - | ns |
| Beta 2 microglobulin, mg/L | 127 | 97% | 3.8 | 2.75-6.15 | 166 | 98% | 3.75 | 2.48-5.2 | 212 | 100% | 3.27 | 2.4-5 | ns |
| Beta 2 microglobulin <3.5 mg/L | 60 | 47% | - | - | 79 | 47% | - | - | 116 | 55% | - | - | ns |
| Beta 2 microglobulin >5.5 mg/L | 37 | 29% | - | - | 38 | 23% | - | - | 42 | 20% | - | - | ns |
| Albumine, g/dL | 128 | 98% | 3.63 | 3.1-4.03 | 169 | 99% | 3.9 | 3.55-4.2 | 207 | 98% | 3.9 | 3.4-4.2 | 0.0004 |
| Albumine <3.5 g/dL | 52 | 41% | - | - | 40 | 24% | - | - | 55 | 27% | - | - | 0.0044 |
| Creatinine, mg/dL | 128 | 98% | 0.9 | 0.71-1.192 | 170 | 100% | 0.9 | 0.77-1.15 | 211 | 99% | 0.9 | 0.75-1.1 | ns |
| Haemoglobin, g/dL | 130 | 99% | 10.1 | 9.005-11.4 | 170 | 100% | 10.95 | 9.6-12.38 | 212 | 100% | 11.3 | 9.9-12.6 | <0.0001 |
| Haemoglobin <10.5 g/dL | 75 | 58% | - | - | 73 | 43% | - | - | 70 | 33% | - | - | <0.0001 |
| Platelet count, 10^3/mL | 130 | 99% | 200.5 | 159.25-244.5 | 170 | 100% | 229.5 | 187-291.8 | 211 | 99% | 242 | 187-298 | <0.0001 |
| Platelet count <150 10^3/mL | 30 | 23% | - | - | 21 | 12% | - | - | 18 | 9% | - | - | 0.0009 |
| Lactate dehydrogenase, g/dL | 122 | 93% | 450 | 225-480 | 159 | 94% | 312 | 225-477 | 195 | 92% | 441 | 225-480 | ns |
| LDH, Upper Limit | 44 | 48% | - | - | 56 | 44% | - | - | 55 | 38% | - | - | ns |
| Bone marrow plasma cells >60% | 74 | 60% | - | - | 83 | 52% | - | - | 111 | 56% | - | - | ns |
| IG Isotype IgG | 71 | 57% | - | - | 89 | 56% | - | - | 146 | 70% | - | - | 0.008 |
| IG Isotype IgA | 36 | 29% | - | - | 34 | 22% | - | - | 30 | 14% | - | - | 0.006 |
| IG Isotype BJ | 16 | 13% | - | - | 30 | 19% | - | - | 32 | 15% | - | - | ns |
| Light Chain Kappa | 22 | 65% | - | - | 22 | 52% | - | - | 37 | 69% | - | - | ns |
| Light Chain Lambda | 12 | 35% | - | - | 20 | 48% | - | - | 17 | 31% | - | - | ns |
| ISS I | 37 | 29% | - | - | 62 | 37% | - | - | 93 | 44% | - | - | 0.02 |
| ISS II | 50 | 39% | - | - | 65 | 39% | - | - | 76 | 36% | - | - | ns |
| ISS III | 41 | 32% | - | - | 39 | 23% | - | - | 43 | 20% | - | - | 0.05 |
| R-ISS I | 6 | 6% | - | - | 24 | 18% | - | - | 34 | 22% | - | - | 0.0009 |
| R-ISS II | 72 | 69% | - | - | 90 | 69% | - | - | 107 | 68% | - | - | ns |
| R-ISS III | 27 | 26% | - | - | 16 | 12% | - | - | 16 | 10% | - | - | 0.002 |
| t(4,14) | 43 | 34% | - | - | 24 | 14% | - | - | 23 | 12% | - | - | <0.0001 |
| t(6,14) | 1 | 1% | - | - | 1 | 1% | - | - | 1 | 1% | - | - | ns |
| t(11,14) | 11 | 9% | - | - | 36 | 26% | - | - | 34 | 21% | - | - | 0.0016 |
| t(14,16) | 14 | 12% | - | - | 6 | 4% | - | - | 2 | 1% | - | - | 0.0003 |
| t(14,20) | 4 | 3% | - | - | 2 | 1% | - | - | 1 | 1% | - | - | ns |
| FISH Deletion 13 | 111 | 92% | - | - | 97 | 65% | - | - | 14 | 7% | - | - | <0.0001 |
| FISH Deletion 17p | 22 | 17% | - | - | 20 | 13% | - | - | 14 | 7% | - | - | 0.02 |
| FISH Deletion 1p36 | 25 | 23% | - | - | 14 | 10% | - | - | 15 | 9% | - | - | 0.004 |
| FISH Amplification 1q | 109 | 91% | - | - | 50 | 34% | - | - | 9 | 5% | - | - | <0.0001 |
| FISH Hyperdiploidy | 37 | 35% | - | - | 48 | 34% | - | - | 93 | 60% | - | - | <0.0001 |
| Induction (PI) | 102 | 78% | - | - | 131 | 77% | - | - | 159 | 75% | - | - | ns |
| Induction (Imid) | 5 | 4% | - | - | 11 | 6% | - | - | 18 | 8% | - | - | ns |

**Table 2 (continued) | All the clinical and baseline variables available for describing the "BO dataset" were included in this study**

| variable | 1q&13+ (n) | n/N | Median | IQR | 1q/13 (n) | n/N | Median | IQR | 1q&13- (n) | n/N | Median | IQR | P value |
|---|---|---|---|---|---|---|---|---|---|---|---|---|---|
| Induction (PI - Imid) | 22 | 17% | - | - | 27 | 16% | - | - | 33 | 16% | - | - | ns |
| Induction (Other) | 1 | 1% | - | - | 1 | 1% | - | - | 2 | 1% | - | - | ns |
| ASCT | 70 | 62% | - | - | 116 | 72% | - | - | 128 | 65% | - | - | ns |
| Single ASCT | 32 | 28% | - | - | 46 | 28% | - | - | 60 | 30% | - | - | ns |
| Double ASCT | 38 | 34% | - | - | 70 | 43% | - | - | 68 | 35% | - | - | ns |
| Maintenance | 79 | 73% | - | - | 123 | 80% | - | - | 146 | 78% | - | - | ns |
| Consolidation | 42 | 39% | - | - | 75 | 49% | - | - | 84 | 45% | - | - | ns |

The number and percentage of patients' data available for any given variable, along with the median value and inter-quartile range (IQR) are shown here, broken down for each of the three subgroups of the 1q&13 classification that compose the dataset ("1q&13 + ", "1q/13" and "1q&13-"). For categorical variables, two-sided Fisher's Exact test p values were computed, and for continuous variables, two-sided Wilcoxon-Mann-Whitney test p values were computed.

transcriptomic profile, characterized by the over-expression of *CCND2* and whose biomarker is the co-occurrence of gain 1q and loss 13q. Patients carrying this genomic configuration have a dismal prognosis, independently from the presence of other well-known MM prognostic factors, that is worsened when also t(4;14)(p16;q32) is present in the prevalent clone. The classification of patients according to genomic events, possibly driving the pathogenesis of the disease, helps to identify homogeneous MM patients' sub-populations possibly favouring the design of biology-adapted therapeutic treatments.

## Methods
### Patients
The study included 513 newly diagnosed MM patients, with CD138+ cell fractions available at the time of diagnosis. The cohort included three subgroups of patients with different median follow-up (Table 1), including 370, 70 and 73 patients, respectively, upfront treated with therapies including either an immunomodulator (IMiD) or a proteasome inhibitor (PI) or all these agents as part of induction and consolidation treatment followed by IMiD maintenance or no exposure to any continuous therapy.

Patients were either previously enrolled in the EMN02 or in BO2005 clinical trials, whose results has been previously reported[26,27], or consecutively treated in our Institution in the context of the daily clinical practice. The study was approved by Area Vasta Emilia Centro ethics review board (17/2015/U/Tess and 149/2018/Sper/AOUBo) and complied with all relevant ethical regulations. Samples and data were obtained and managed in according with the Declaration of Helsinki. All patients provided signed consent for the genomic analyses. Median progression free survival (PFS) and overall survival (OS) with respective Inter Quartile Range (IQR) were expressed in months. For the whole cohort of patients, they were respectively 43 (IQR:17-67) and 63 months (IQR: 31-78). Patient baseline clinical characteristics are summarized in Table 1. We declare that no gender-based analysis was carried out as this covariate is not known to impact the biology of Multiple Myeloma and data was not collected. On the contrary, sex and age data were collected and described in Tables 1, 2. Accordingly, sex and age covariates were considered in the clinical analyses, univariate and multivariate survival models.

### MMRF CoMMpass validation cohort
All the raw data coming from the MMRF CoMMpass study (NCT01454297) and employed as validation cohort in this study, are publicly available[44]. The study is ongoing, with data released regularly for research use via the MMRF research gateway, https://research.themmrf.org[45]. In this study, we used Interim Analysis (IA) 13.

Somatic CN profiles for the definition of CNAs in MM were generated from 752 NDMM patients from the CoMMpass study, by low coverage long-insert WGS (median 4-8x). Gene expression data, as obtained by RNA-seq gene counts, were downloaded for 659 newly diagnosed patients.

### SNPs arrays experiments and analysis pipeline
Total genomic DNA was isolated using Maxwell® 16 LEV Blood DNA kit (Promega, Madison, WI). SNP array profile experiments were carried out according to the manufacturer's protocols (Affymetrix SNP6.0 and Cytoscan HD Genome-wide Human GeneChip, Affymetrix, Santa Clara, CA). For each patient, SNPs array raw data were analysed with a bioinformatic pipeline including RawCopy v1.1[46], ASCAT 2.5.2[47] and Broad Institute GISTIC 2.0[48] algorithms, aimed at calling and mapping clonal CNAs across the whole genome.

The genomic segments profiles were generated using RawCopy R package and PSCBS algorithm. The significance threshold for segmentation was set at $10^{-7}$. RawCopy was employed to normalize Affymetrix arrays, extract quality metrics and finally obtain raw logR and BAF tracks. Quality metrics were also produced by Chromosome

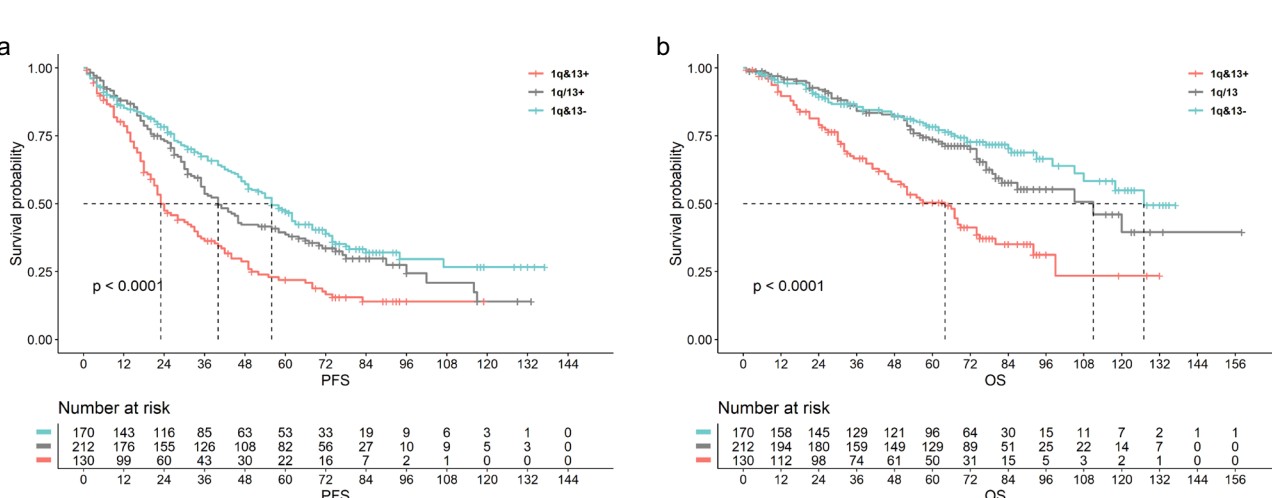

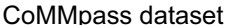

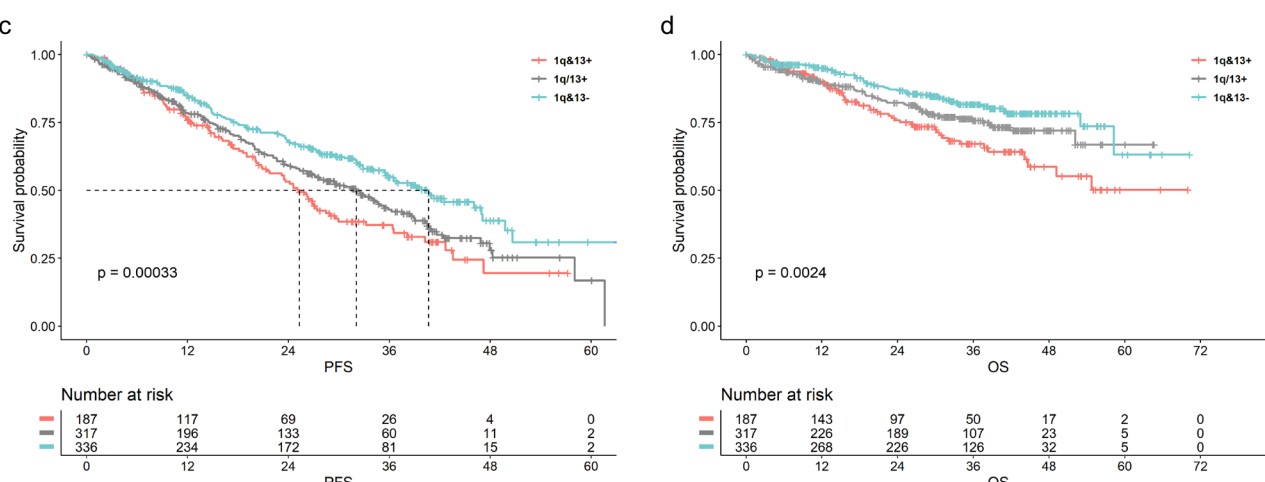

**Fig. 7 | Clinical impact of 1q&13 classification.** Effect on PFS and OS of 1q&13 classification: **a** BO dataset (*n* = 512 patients) 5-year PFS: 22% vs. 37% vs. 47%, log-rank test *p* = 0.000042; (**b**) OS: 50% vs. 74% vs. 78%, log-rank test *p* = 0.0003, for 1q&13 + , 1q/13, 1q&13-, respectively. **c** CoMMpass dataset (*n* = 752 patients) 3-year PFS: 37% vs. 43% vs. 55%, log-rank test *p* = 0.0003; (**d**) OS: 67% vs. 76% vs. 81%, log-rank test p = 0.002, for 1q&13 + , 1q/13, 1q&13-, respectively.

Analysis Suite (ChAS) v3.3 analysis. All samples passed quality thresholds defined as RawCopy MAPD < 0.23, ChAS MAPD < 0.25 and ChAS QC < 10.00. The raw logR and BAF tracks of all samples that passed the quality checks were used as input for ASCAT 2.5.2. This analysis produced a genomic CN track *per* patient, adjusted and corrected for its relative computed normal cell contamination level. This step removes the effect of imperfect enrichment of tumour samples, enabling the quantification of sub-clonal CNAs. According to the ASCAT computed samples' purity, a Cancer Cell Fraction (CCF) was defined for each alteration, spanning from 0% to 100%. All ASCAT purity solutions were manually reviewed and samples presenting an ambiguous ASCAT ploidy or purity solution, possibly reflecting a whole genome doubling karyotype, were refitted to match a diploid state. Segments raw Log2 ratio tracks were subsequently converted to CN values using the formula described in Van Loo et al.[47]. The copy number thresholds for single copy gain and single copy loss were set at 2.1 CN and 1.9 CN, respectively. The CN thresholds for two or more copy gain and homozygous loss were set at 3.4 and 0.6, respectively. The B Allele Frequency threshold used to detect presence of LOH events was set at 0.8. GISTIC 2.0 tool was employed to detect both "broad" arm-level

CNAs (defined as alteration spanning >25% of the chromosome arm) and significative focal CNAs regions. A complete call-set for each sample and each chromosome arm was built, keeping in consideration both broad arm-level CNAs calls and any CNA detected in a focal region. In particular, a set of 8 focal genomic regions, with a non-random confluence of highly frequent, small CNAs, covering well-known tumour suppressor genes and oncogenes, widely regarded as relevant in MM biology (*TP53, RB1, MYC, CKS1B, ANP32E/MCL1, FAM46C/CDKN2C/FAF1, TRAF3, CYLD*) was included. (Supplementary Fig. S1). Since we detected various levels of sub-clonality in the total CNAs call-set, we decided to put a threshold on CN calls in order to select only the CNAs that were present in at least 50% of tumour cells (the majority of cells that compose a tumour sub-clonal architecture). The threshold was set up at 50% also because this level represents both empirically and visually a good separation between CNAs with a high or low clonality level.

**Co-occurrence analysis**
To investigate the relationships between different genomic alterations in our dataset, we constructed a correlation matrix, in which each

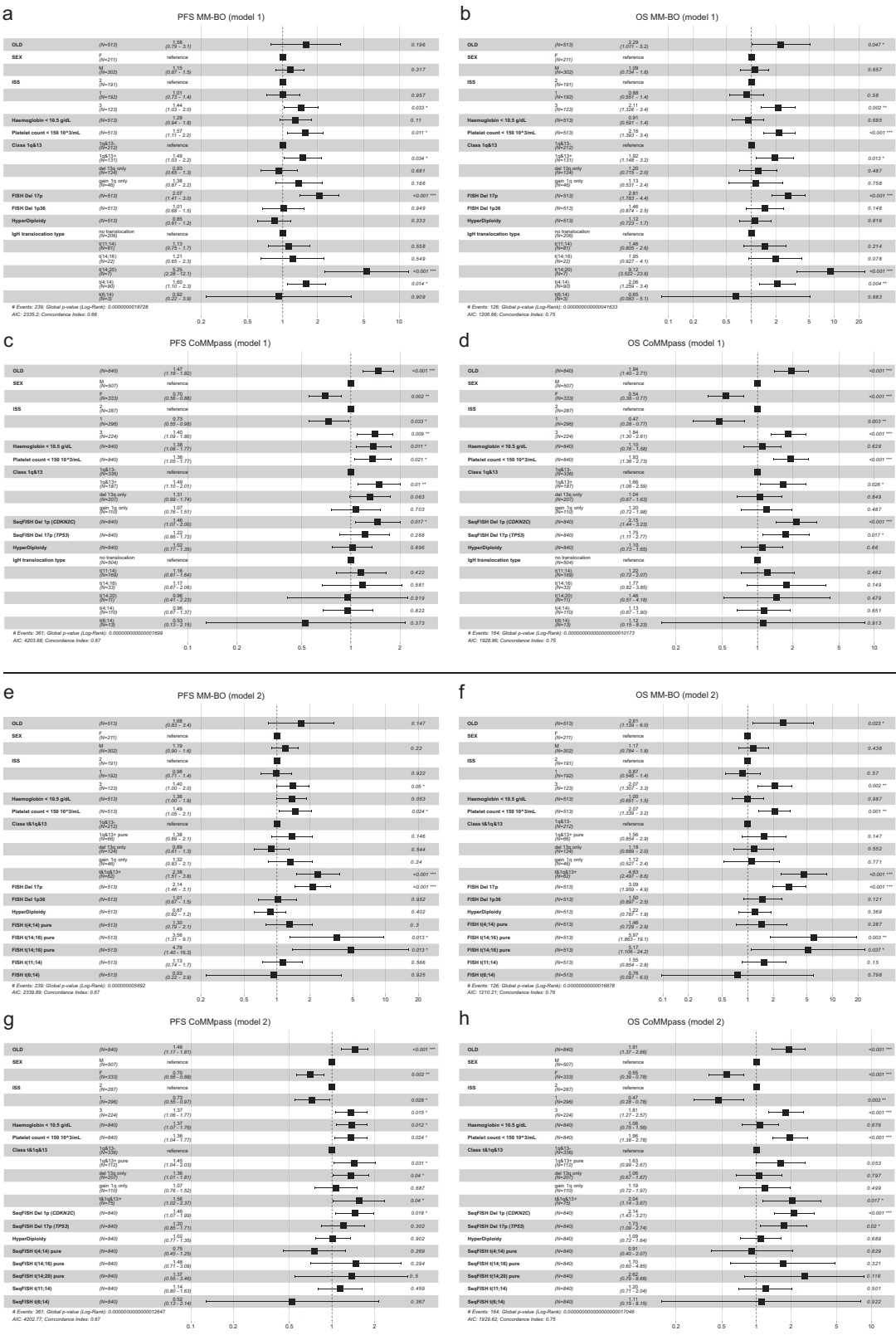

**Fig. 8 | Multivariate models for 1q&13 classification.** Forest plots of PFS and OS on Cox multivariate analyses: (**a**–**d**) model 1 (upper four panels) explored the risk of "1q&13" genomic configuration classes, (**e**–**h**) model 2 (lower four panels) explored the "t&1q&13" risk category classes. Analyses were performed both in the BO–and

in the CoMMpass datasets. The squares along the centre line represent point estimates of the Hazard Ratios (HR) of variables included in the models, while horizontal lines extending from the squares represent 95% confidence intervals for each variable's HR.

element represents the Pearson's correlation coefficient between a pair of genomic alterations. These alterations were measured as continuous data, with clonality values ranging between 50% and 100%.

In addition, we also constructed a Jaccard similarity matrix. This matrix was used to measure the intersections between pairs of genomic alterations, which were treated as dichotomized data in this context.

In our analysis, we presented multiple direct correlations independently from one another. Each genomic alteration was considered as a separate phenomenon, not contingent on the others. This approach allowed us to highlight each distinct biological correlation that we were investigating independently. Given the independent nature of our hypotheses, we chose not to apply a correction for multiple testing to avoid an unnecessary reduction in statistical power and the potential overlooking of meaningful relationships.

### Gene expression profiling pipeline

We performed three different comparisons of Differentially Expressed Genes (DEGs) ("1q&13+ vs 1q&13-", "1q&13+ vs 1q/13", "1q&13- vs 1q/13"), to compare all the possible 1q&13 group combinations, starting from the RNA-sequencing (RNA-seq) data of 659 patients, as downloaded from CoMMpass study. The same pipeline, including Limma, Glimma and edgeR R packages[49], was applied to all groups' comparisons. We used edgeR package to keep only those genes presenting a Count Per Million (CPM) > 1 in at least $n = 5$ samples and for the *per*-sample library size normalization, using the trimmed mean of M values method (TMM). Following, the "voom" method, linear modelling, and empirical Bayes moderation of the Limma package were used to adjust the variance and perform gene set testing. Finally, dimensionality reduction analysis (MDS) on gene expression data was performed using Glimma "glMDSPlot" function. We further filtered the DEGs data including only the significative genes with Fold change > 2. By doing so, the only comparison that included a considerable number of DEGs was "1q&13+ vs 1q&13-" ($N = 301$ DEGs, 179 up-regulated, 122 down-regulated) (Supplementary Data S3).

### Gene Set Enrichment Analysis (GSEA)

The starting list of DEGs was used to explore pathways possibly affected by the co-occurrence of 1q CN gain and 13q CN loss. We applied a pre-ranked analysis, after removing the immunoglobulin heavy and light genes to reduce the noisy effect of this PC gene intrinsic expression signal. By setting a fold change cut-off > 1, we therefore included 1564 genes (1002 up-regulated and 562 down-regulated) in the enrichment analysis. GSEA version 4.1.0 software was used to analyze genes function from the GSEA website MSIGDB database (http://software.broadinstitute.org/gsea/msigdb). The default weighted enrichment method was applied for enrichment analysis. The random permutations number was set for 1000 times. The analysis was performed with the following settings: FDR < 0.25, NOM p-value < 0.05 and |NES| > 1.

### Dimensionality reduction

With the aim of creating a label to describe the driver and founding genomic events in MM, we restructured our set of genomic annotations as follows: (1) we first grouped all the amplifications of the odd numbered chromosomes in a label named "HD" (HyperDiploidy), called only if at least 2 or more arms of two different odd chromosomes were amplified; (2) we then grouped all the different, mutually exclusive IgH translocations in a label named "t-IgH", in order to capture the unique common origin of all these driver events.

We filtered out all the CNAs which frequencies were under 5% in the total population, obtaining a list of 67 alterations, employed as input for the dimensionality reduction analysis as binary alteration variables, namely: "Amp 10p", "Amp 10q", "Amp 12p", "Amp 12q", "Amp 13q", "Amp 14q", "Amp 16p", "Amp 16q", "Amp 17p", "Amp 17q", "Amp 18p", "Amp 18q", "Amp 1p", "Amp 1q", "Amp 20p", "Amp 20q", "Amp 22q", "Amp 2p", "Amp 2q", "Amp 4p", "Amp 4q", "Amp 6p", "Amp 6q", "Amp 8p", "Amp 8q", "Del 10p", "Del 10q", "Del 11p", "Del 11q", "Del 12p", "Del 12q", "Del 13q", "Del 14q", "Del 15q", "Del 16p", "Del 16q", "Del 17p", "Del 17q", "Del 18p", "Del 18q", "Del 19p", "Del 19q", "Del 1p", "Del 1q", "Del 20p", "Del 20q", "Del 21q", "Del 22q", "Del 2p", "Del 2q", "Del 3p", "Del 3q", "Del 4p", "Del 4q", "Del 5p", "Del 5q", "Del 6p", "Del 6q", "Del 7p", "Del 7q", "Del 8p", "Del 8q", "Del 9p", "Del 9q", "HD" and "t-IgH".

Given the binary nature of data, to reduce the dimensionality of the dataset, both a Principal Components Analysis (PCA), without centreing nor scaling, and a Non-metric Multi-Dimensional Scaling (NMDS) analysis were performed.

We used "prcomp" function from "Stats" R package to perform PCA, while "metaMDS" function from "Vegan" R package was used to perform NMDS, using "monoMDS" engine and choosing Manhattan distances between observations. We performed both analyses because the NMDS procedure is considered more appropriate for dealing with binary data by some authors[50]. In fact, unlike other ordination techniques that rely on (primarily Euclidean) distances, such as PCA, NMDS uses rank orders providing an extremely flexible technique that can accommodate a variety of different types of data. Thus, we applied both procedures to confirm the consistency of the results, ultimately obtained similar findings.

The stress values in all NMDS analysis with $n = 2$ dimensions were <0.20, while the stress values in all NMDS analysis with $n = 3$ dimensions were <0.13, confirming the acceptable goodness of the solutions reached by the algorithm in reducing the dimensionality of data. The "knee" in the scree plot of the PCA confirmed that $n = 2$ dimensions is the optimal number of dimensions to choose for representing the global complexity of the data.

### Clinical and statistical analysis

All the analyses were conducted using R language and environment for statistical computing (R Foundation for Statistical Computing, Vienna, Austria). The analysis was performed with a significance level of at least 0.05 and all variables objected of inference were reported together with their 95% confidence intervals (CI). The genomic complexity of 1q&13 classification was explored by comparing characteristics between groups with non-parametric methods such as Kruskal-Wallis's test on the medians (or the parametric t-test on means). For the parametric comparisons, the Pearson's test was employed. PFS was measured in months, from the start of therapy to the event of first progression of the disease or death. OS considered death as outcome/event and was measured from the same landmark. Univariate survival analysis on both PFS and OS were performed by the Kaplan-Meier method, as for drawing the survival curves. Semi-parametric Cox regression analysis was adopted to estimate hazard ratios (HR) with an 95% CI between predefined possible prognostic groups. Multivariate analysis was performed again by Cox regression analysis to identify the abnormalities independently affecting the prognosis with their HR and 95% CI, stratifying for sub-group variable in BO dataset, in order to adjust the HR.

### Reporting summary

Further information on research design is available in the Nature Portfolio Reporting Summary linked to this article.

## Data availability

The CoMMpass study publicly available data used in this study are available to download, after requesting and obtaining access from GENOSPACE, from the MMRF research gateway https://research.themmrf.org/[45]. All the data contained in BO dataset, including raw copy number signal SEG files and clinical data, generated in this study have been deposited in the GitHub repository https://doi.org/10.5281/zenodo.10277460.[52]. The processed data used for figures and tables

are also available in the same GitHub repository. The source data (including data used for the creation of figures) are provided with this paper. Source data are provided with this paper.

## Code availability

Results from this paper were generated using the code freely available on the GitHub repository http://github.com/andrea-poletti-unibo/1q-13_paper. https://doi.org/10.5281/zenodo.10277460[51].

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

## Acknowledgements

This work was supported by the Associazione Italiana Ricerca sul Cancro (AIRC) IG22059 and the Italian Health Ministry – Ricerca Finalizzata RF-2016-02362532. C.T. and M.M. are supported by the H2020 European grant GenoMed4ALL (grant n.101017549). E.B., M.M. and C.T. are supported by BolognaAIL (Associazione Italiana contro le leucemie-linfomi e mielomi). A.P. has been supported by the International Myeloma Society Career Development Award 2022.

## Author contributions

C.T. and A.P. contributed equally to the paper; they designed and supervised the study, collected and analyzed data and wrote the paper; V.S. designed the study, collected and analyzed data and wrote the paper; M.M., E.Z. and G.Maz. collected, analyzed and interpreted the data and critically revised the manuscript; E.B. and I.V. collected data and critically revised the manuscript; S.A., L.P., B.T., N.T., G.Mar., A.K., I.P., P.T., K.M., S.R. and I.R. collected data; M.C. critically revised the manuscript and approved the submitted version.

## Competing interests
The authors declare no competing interests.
