## [Peer Review File · Nature Communications]

Multi-dimensional scaling techniques unveiled gain1q&loss13q co-occurrence in Multiple Myeloma patients with specific genomic, transcriptional and adverse clinical features.Reviewers' Comments:

Reviewer #1:

Remarks to the Author:

In this study, Terragna et al. aim at dissecting the genomic landscape of newly diagnosed MM patients, modelling interactions between chromosomal alterations. The strength of the study is the use of a large cohort of clinically annotated newly diagnosed patients. However, majority of the analyses performed required clarifications, and the novelty is low with most of these data/results already published in several other studies.

1. Focusing only on major CNAs (as stated on page 5 117-120) is biased; in CNA context we don't know what is passenger and what is driver.
2. How did they define clonality groups?
3. Figure 2: the analysis is not accurate as it is not appropriate to measure correlation between dichotomized CNAs. A better measure is co-occurrence.
4. The authors describe the dimensional scaling techniques (NMDS and PCA) as being more appropriate to describe and eventually reduce the complexity of the whole dataset of annotated CNAs. More appropriate compared to what?
5. Figure 3 is confusing, and it is hard to understand what the data represents.
6. The long description in pages 13 and 14 regards data that were all published before.

Reviewer #2:

Remarks to the Author:

This manuscript provides evidence of differences in gene expression and survival that suggest that amp1q/del13p characterizes a distinct subtype of MM at high risk. A claim is made that Commonly employed MM risk models do not precisely partition high- from low-risk patients. However, uncited previous research has demonstrated divisions in risk among MM patients [1–6]. Furthermore, the emphasis on a multidimensional clustering approaches seems misplaced, since it's unclear how statistically meaningful the clusters are—especially given that the clustering input effectively presupposed the clusters (amp1q [2,3], odd-chromosome trisomies, del13q, tIGH); evolutionary [7] and other bioinformatic approaches [8,9] may be more insightful. In general, the research needs comprehensive contextualization justifying importance [10–12], the writing needs improvement, the paper could be significantly shorter, and more clearly organized.

Line comments

38: Abstract: "This allowed to": for clarity, please add a noun after "This".

53: "Very effective therapeutic protocols" may be an overstatement. 5-year survival of roughly 50–60% might be substantially better than previous options, but may not be considered "very effective". The authors go on to note that many patients don't respond to current treatment, and they could consider introducing this information earlier.

62: The concept of "odd-numbered chromosomes trisomies" is biologically confusing. If it's simply a mnemonic that the common MM trisomies are in certain odd-numbered chromosomes, perhaps it would be best to list the specific chromosomes.

72–74: Vague; this section should be more specific about what previous studies have found.

75–85: This paragraph seems unnecessary—the idea that the genomic features of MM can be used to identify subgroups could be relayed in a single sentence elsewhere.

93–100: These are results/discussion, not background.

109: The phrasing here is odd, because it suggests that the use of an alternative (and, for the purposes of CNA and t-IgH detection, inferior) technology in the validation data set is a benefit, when the benefit is simply the use of an independent validation data set to test the findings from the training data set.

125: Does this measurement cover all odd-numbered chromosomes? How does the even-chromosome measurement compare?

163–201: Significant correlations should be presented more clearly, and in the context of correction for multiple testing.

170: Rare events do not necessarily contribute less to pathogenesis [13]. Furthermore, as odd-numbered chromosome gains are lumped together in this analysis, it's unsurprising that they collectively appear at higher frequencies than other CNAs considered.

192: The resolution of figure 2 is too low to read the figure.

221: Why, in line 61 of Supplementary Methods, are t_IgH events listed as being filtered out due to low prevalence, when here in the main manuscript they are described as being merged together?

247–254: Panels A and B should be described in the figure description. On line 253, the meaning of "resumed the clusters' position" is not clear. How are the specific points depicted determined?

263–266: Neither assumption of ancestry seems valid. A more prevalent genomic trait does not necessarily occur earlier than other traits—it may simply occur and reach high allele frequency (due to a combination of high mutation rates and/or strong selection [13]) more often than less common variants. Similarly, association with a "simple genetic background" may indicate that the trait substantially contributes to MM without need for additional complex genomic alterations. However, both prevalence and lack of other explanatory events contribute to the argument for the importance of the variants, so a different framing in lieu of ancestry could be more convincing.

286: Does "emersion" mean emergence here?

339–340: For clarity, please explain here why excluding t-IgH patients serves to confirm the DEG pattern. It becomes clear later in the manuscript.

388: The notation is somewhat confusing: 1q&13+ means positive for gain of 1q and loss 13q, while 1q&13- means neutral copy number in both. Consider if there is another way to annotate the groups.

1. Ortiz-Estévez M, Towfic F, Flynt E, Stong N, Jang IS, Wang K, et al. Integrative multi-omics identifies high risk multiple myeloma subgroup associated with significant DNA loss and dysregulated DNA repair and cell cycle pathways. *BMC Med. Genomics* 2021;14:295.
2. Hanamura I. Gain/Amplification of Chromosome Arm 1q21 in Multiple Myeloma. *Cancers* 2021;13:256.
3. Sklavenitis-Pistofidis R, Getz G, Ghobrial I, Papaioannou M. Multiple Myeloma With Amplification of Chr1q: Therapeutic Opportunity and Challenges. *Front. Oncol.* 2022;12:961421.
4. Flietner E, Yu M, Poudel G, Veltri AJ, Zhou Y, Rajagopalan A, et al. Molecular characterization stratifies VQ myeloma cells into two clusters with distinct risk signatures and drug responses. *Oncogene* 2023;42:1751–62.

5. Bolli N, Biancon G, Moarii M, Gimondi S, Li Y, de Philippis C, et al. Analysis of the genomic landscape of multiple myeloma highlights novel prognostic markers and disease subgroups. *Leukemia* 2018;32:2604–16.
6. Cardona-Benavides IJ, de Ramón C, Gutiérrez NC. Genetic Abnormalities in Multiple Myeloma: Prognostic and Therapeutic Implications. *Cells* [Internet] 2021;10. Available from: <http://dx.doi.org/10.3390/cells10020336>
7. Maura F, Landgren O, Morgan GJ. Designing Evolutionary-based Interception Strategies to Block the Transition from Precursor Phases to Multiple Myeloma. *Clin. Cancer Res.* 2021;27:15–23.
8. Dao P, Kim YA, Wojtowicz D, Madan S, Sharan R, Przytycka TM. BeWith: A Between-Within method to discover relationships between cancer modules via integrated analysis of mutual exclusivity, co-occurrence and functional interactions. *PLoS Comput. Biol.* 2017;13:e1005695.
9. Kim YA, Cho DY, Dao P, Przytycka TM. MEMCover: integrated analysis of mutual exclusivity and functional network reveals dysregulated pathways across multiple cancer types. *Bioinformatics* 2015;31:i284–92.
10. Castaneda O, Baz R. Multiple Myeloma Genomics - A Concise Review. *Acta Med. Acad.* 2019;48:57–67.
11. Nishihori T, Shain K. Insights on Genomic and Molecular Alterations in Multiple Myeloma and Their Incorporation towards Risk-Adapted Treatment Strategy: Concise Clinical Review. *Int. J. Genomics Proteomics* 2017;2017:6934183.
12. Klein MI, Cannataro VL, Townsend JP, Newman S, Stern DF, Zhao H. Identifying modules of cooperating cancer drivers. *Mol. Syst. Biol.* 2021;17:e9810.
13. Cannataro VL, Gaffney SG, Townsend JP. Effect Sizes of Somatic Mutations in Cancer. *J. Natl. Cancer Inst.* 2018;110:1171–7.

Reviewer #3:

Remarks to the Author:

Rebuttal Letter - NCOMMS-23-18320

Reviewer #1

(Remarks to the Author): with expertise in multiple myeloma, genomics.

In this study, Terragna et al. aim at dissecting the genomic landscape of newly diagnosed MM patients, modelling interactions between chromosomal alterations. The strength of the study is the use of a large cohort of clinically annotated newly diagnosed patients. However, majority of the analyses performed required clarifications, and the novelty is low with most of these data/results already published in several other studies.

1. Focusing only on major CNAs (as stated on page 5 117-120) is biased; in CNA context we don't know what is passenger and what is driver.

Thank you for your insightful comment regarding our focus on major CNAs. We understand your concern about the potential bias in distinguishing between driver and passenger CNAs. In our study, we focused on major CNAs due to their prevalent distribution (Fig.1a, b) and to remove technical signal noise biases potentially affecting minor-subclonal alterations.

However, we agree that the role of minor CNAs should not be overlooked as they could also play significant roles in disease progression and patient prognosis.

To address your concern, we rephased accordingly our paper (Lines 117-119) acknowledging the reason of our choice, and removing the mention to clinical relevance of clonal/sub-clonal alterations.

We believe that this choice will provide a clearer view of the genomic landscape in multiple myeloma and will address the concern you raised. Thank you again for your valuable feedback.

2. How did they define clonality groups?

In our research, we defined clonality groups based on the quantitative estimation of chromosomal alterations. We categorized these alterations as "clonal" if they affected virtually all (>90%) tumor cells, "sub-clonal-major" if they affected the majority (between 50% and 90%) of tumor cells, and "sub-clonal-minor" if they affected the minority (between 10% and 50%) of tumor cells (Page 7, Lines 114-117).

In particular, 90% and 10% cutoffs were chosen based on commonly reported error of CytoScan HD array measures (± 0.10 CN units)¹.

1) Scioni, F. et al. (2018). The cytoscan HD array in the diagnosis of neurodevelopmental disorders. High-throughput, 7(3), 28. doi: 10.3390/ht7030028

3. Figure 2: the analysis is not accurate as it is not appropriate to measure correlation between dichotomized CNAs. A better measure is co-occurrence.

Thank you for your insightful comment regarding our use of correlation to analyze dichotomized CNAs. We understand your concern about the appropriateness of this method and appreciate your suggestion to use co-occurrence as a measure.

In response to your comment, we revised our analysis to measure the co-occurrence of CNAs by using a Jaccard similarity matrix, added as supplementary figure S2, and described in the main text (Lines 187-190). However, we believe that the Pearson matrix, while not appropriate on binary data, is valuable in defining pattern of both negative and positive correlations, which is something that a co-occurrence index is not able to capture, but is crucial for the chapter overall

message. Consequently, we modified our Pearson correlation matrix by changing the input alterations data from binary to continuous variables (in a 50-100% clonality range, as defined in the new figure caption. Lines 203-204), and we changed the figure accordingly. Notably this change does not affect the overall message of this chapter.

We believe that this modification will provide a more accurate and meaningful interpretation of our data.

4. The authors describe the dimensional scaling techniques (NMDS and PCA) as being more appropriate to describe and eventually reduce the complexity of the whole dataset of annotated CNAs. More appropriate compared to what?

We apologize if our original phrasing led to confusion. We did not intend to suggest that these techniques are more appropriate compared to other specific methods. Rather, our intention was to convey that these techniques are particularly suited for dealing with high-dimensional datasets, such as the dataset of annotated CNAs in our study. The referee was corrected.

5. Figure 3 is confusing, and it is hard to understand what the data represents.

Figure 3 was substantially improved to be more intuitive and easier to understand.

In details:

- Labels describing the various sub-figures were added
- Legends were improved. Every plot has now its own legend or description.
- The panels and the figures' letters were rearranged in order to highlight the difference between BO-MM and CoMMpass datasets
- Solid lines that connect the cluster centroids in the bottom figures were replaced by dashed lines in order to not being confused with the NMDS axis.
- The figure caption was updated and improved to reflect those changes.

The manuscript text was modified accordingly (Lines 230-238).

6. The long description in pages 13 and 14 regards data that were all published before.

The chapter was substantially rephrased and changed, also based on suggestion of reviewer #2. Now the description is shorter and straight to the point. The "timing/ancestrality" interpretation of the score was dropped in favor of a more appropriate "driverness" score interpretation. Importantly the score was not changed, but only its interpretation and description. We believe that the new description of our unique analysis can now highlight the difference between other already published MM timing papers and our results.

Reviewer #2

(Remarks to the Author): with expertise in bioinformatics, biostatistics, genomics

This manuscript provides evidence of differences in gene expression and survival that suggest that amp1q/del13p characterizes a distinct subtype of MM at high risk. A claim is made that Commonly employed MM risk models do not precisely partition high- from low-risk patients.

However, uncited previous research has demonstrated divisions in risk among MM patients [1–6].

The commonly used definition of high-risk multiple myeloma (MM) often neglects the consideration of disease biology. Instead, most commonly employed risk scores are based primarily on statistical factors. As a result, although previous research has demonstrated the clinical benefits of using different risk scores, none has provided a comprehensive understanding of the underlying biology behind high-risk disease. We thank for the suggested papers to cite; among them, we will add the following: Bolli N, Biancon G, Moarii M, Gimondi S, Li Y, de Philippis C, et al. Analysis of the genomic landscape of multiple myeloma highlights novel prognostic markers and disease subgroups. *Leukemia* 2018;32:2604–16 (line 79).

Furthermore, the emphasis on a multidimensional clustering approaches seems misplaced, since it's unclear how statistically meaningful the clusters are—especially given that the clustering input effectively presupposed the clusters (amp1q [2,3], odd-chromosome trisomies, del13q, tIgH); evolutionary [7] and other bioinformatic approaches [8,9] may be more insightful.

In our paper it has not been possible to use the evolutionary approach employed in ref. 7 (Maura et al. 2021) or in other famous reconstructions of tumor life histories papers^{1,2}, since phylogenetic trees reconstructions or signatures analyses included in those papers are strictly based on mutational alterations data, derived from sequencing experiments. On the contrary our main study cohort BO-MM only includes SNP array CN data and FISH t-IgH data, which present a fundamentally different data structure and format if compared to mutation data.

The “BeWith” bioinformatic method in ref. 8 suffers the same issue, since it requires, and it's designed to be used on mutations alterations data, which are way more abundant than CN/SV alterations. In fact, one typical MM sample presents several hundreds of mutations but only tens of CNAs and one t-IgH. Consequently we think that our dataset is underpowered to be analyzed effectively with such bioinformatic method.

Finally, also the subnetworks analysis by MEMCover in ref.9 strictly requires input mutation data, as stated in the paper and in the GitHub repository README file³.

Therefore, while we agree that evolutionary and other bioinformatic approaches may be highly insightful, unfortunately they are not applicable in our case.

In conclusion, since our paper is based on CNAs analysis we believe that the NMDS approach is very well suited to be applied in this setting.

1) Gerstung, M., et al. (2020). The evolutionary history of 2,658 cancers. *Nature*, 578(7793), 122-128.

2) Nik-Zainal, S., et al. (2012). The life history of 21 breast cancers. *Cell*, 149(5), 994-1007.

3) <https://github.com/yooah/MEMCover>

In general, the research needs comprehensive contextualization justifying importance [10–12], the writing needs improvement, the paper could be significantly shorter, and more clearly organized.

Line comments

38: Abstract: “This allowed to”: for clarity, please add a noun after “This”.

Thank you for the remark, the inaccuracy has been corrected

53: “Very effective therapeutic protocols” may be an overstatement. 5-year survival of roughly 50–60% might be substantially better than previous options, but may not be considered “very effective”. The authors go on to note that many patients don’t respond to current treatment, and they could consider introducing this information earlier.

While it may be accepted to remove the term "very" from the sentence to moderate the portrayal of clinical results in multiple myeloma (MM) disease outcomes, it is still undeniable that there has been a notable and objective improvement in the survival of MM patients over the last five years. This achievement is widely recognized within the MM scientific community, affirming the significant progress made in this field.

62: The concept of “odd-numbered chromosomes trisomies” is biologically confusing. If it’s simply a mnemonic that the common MM trisomies are in certain odd-numbered chromosomes, perhaps it would be best to list the specific chromosomes.

Thank you for your comment regarding the concept of "odd-numbered chromosomes trisomies". We appreciate your suggestion to list the specific chromosomes involved.

However, we would like to clarify that the term "odd-numbered chromosomes trisomies" is not merely a mnemonic, but rather a recognized entity in the context of multiple myeloma (MM). It is well-documented in the literature that trisomies in MM predominantly occur in odd-numbered chromosomes, specifically chromosomes 3, 5, 7, 9, 11, 15, 19, and 21. This pattern is a distinctive feature of hyperdiploid MM, one of the two major subtypes of MM ¹.

We understand that the term may be confusing for readers unfamiliar with this aspect of MM biology. To address this, we propose to include in the bibliography the above-mentioned reference (line 64).

72–74: Vague; this section should be more specific about what previous studies have found.

Thank you for your comment. We added the following explanation, in order to better clarify the message: “even when mutation and gene fusions data were included in group clustering. In fact the final MM groupings defined in those papers were defined mostly by CNA and t-IgH.” (lines 74-76).

75–85: This paragraph seems unnecessary—the idea that the genomic features of MM can be used to identify subgroups could be relayed in a single sentence elsewhere.

Thank you for your feedback regarding the paragraph in lines 75-85. We understand your point about the potential redundancy of this information.

However, we believe that this paragraph is necessary for our manuscript. The purpose of this paragraph is to emphasize the importance and novelty of our approach, which is to use the intrinsic biology of MM to identify clinically relevant subgroups. This is in contrast to other biological classifications and scoring systems, mainly based on statistical considerations, which may not fully capture the biological complexity of MM.

We believe that this paragraph provides important context and justification for our study, since it highlights the original contribution of our research in the context of existing literature.

93–100: These are results/discussion, not background.

Thank you for your feedback regarding the placement of the information in lines 93-100. We understand your point that these lines seem to be more aligned with the results/discussion section rather than the background.

However, it is common practice in scientific writing to include a brief overview of the main findings in the introduction section (for example in the introductions of MEMCover paper and BeWith paper that you cited, a brief overview of the results are provided, deferring the detailed explanation in the results sections).

We believe that this serves to provide readers with a quick summary of the results and their significance, setting the context for the detailed results and discussion that follow. In our manuscript, the information in lines 93-100 was intended to serve this purpose.

109: The phrasing here is odd, because it suggests that the use of an alternative (and, for the purposes of CNA and t-IgH detection, inferior) technology in the validation data set is a benefit, when the benefit is simply the use of an independent validation data set to test the findings from the training data set.

Thank you for your feedback regarding the phrasing in line 109. We understand your concern about the potential misinterpretation that the use of an alternative technology for CNA and t-IgH detection in the validation data set is a benefit.

We agree that the primary benefit is indeed the use of an independent validation data set to test the findings from the training data set. The use of an alternative technology was not intended to be presented as a benefit, but rather as a methodological detail.

In response to your comment, we propose to revise the phrasing in line 109 to clarify this point. We could rephrase it as follows:

“We further validated our findings using this independent data set, which also provided an opportunity to test the robustness of our results by using a different detection technology”.

We believe this revision will more accurately convey the intended message and avoid potential confusion.

125: Does this measurement cover all odd-numbered chromosomes? How does the even-chromosome measurement compare?

Thank you for your question regarding our measurement of odd-numbered chromosomes trisomies.

In our study, we observed that the highest CN gains were predominantly found in odd-numbered chromosomes, specifically chromosomes 3, 5, 7, 9, 11, 15, 19, and 21. Odd-numbered chromosomes CN gains involved preferentially chromosomes 19 (46.1%), 9 (45.2%), 15 (43.8%), 11 (38.1%), 5 (37.9%), 3 (36.5%), 7 (28.8%), 21 (22.6%), as ranked by their frequencies. (Page 6, Lines 132-135)

This pattern is a distinctive feature of hyperdiploid MM, one of the two major subtypes of MM. As for even-numbered chromosomes, our study does not provide a direct comparison, since even-numbered chromosomes gains are not a distinctive feature of MM.

163–201: Significant correlations should be presented more clearly, and in the context of correction for multiple testing.

Thank you for your feedback regarding the presentation of significant correlations in lines 163-201. We understand your concern about the clarity of these results and the need for correction for multiple testing.

In our study, our aim was to present multiple direct comparisons independently from one another. Each comparison was considered as a separate analysis not contingent on the others. Therefore, while we understand the common practice of applying a correction for multiple testing when exploring multiple comparisons within the same dataset, in this specific context, we did not apply such a correction.

The rationale for this approach is as follows: Our study is focused on the exploration of specific genomic alterations and their co-occurrence in multiple myeloma. Each of these alterations and their potential relationships represent distinct biological phenomena that we are investigating independently. As such, the comparisons we are testing are not multiple comparisons of a single hypothesis, but rather separate hypotheses in their own right. Therefore, we believe that applying a correction for multiple testing in this context could unnecessarily reduce the statistical power and potentially overlook meaningful relationships.

We appreciate the importance of clearly stating our methodology and discussing the potential implications for the interpretation of our results. In response to your comment, we propose to explicitly state our approach to multiple testing in the methods sections. (Lines 650-664)

Additionally, according to the comment of reviewer 1, we decided to implement this analysis with the addition of co-occurrence measures (Jaccard index matrices, supplementary figure), which is more appropriate when dealing with binary data.

170: Rare events do not necessarily contribute less to pathogenesis [13]. Furthermore, as odd-numbered chromosome gains are lumped together in this analysis, it's unsurprising that they collectively appear at higher frequencies than other CNAs considered.

Thank you for your comment regarding line 170. We understand your point about the potential significance of rare events in pathogenesis.

In our manuscript, the statement in line 170 was intended to refer to the complexity of the genomic landscape in multiple myeloma, rather than the contribution of specific events to pathogenesis. We agree that rare events can indeed play significant roles in pathogenesis, and we did not mean to suggest otherwise.

192: The resolution of figure 2 is too low to read the figure.

Thank you for bringing to our attention the issue with the resolution of Figure 2. We completely changed and revised Figure 2 to improve its resolution.

Of note, we would like to pinpoint that the figures included in the manuscript were added for ease of the reader. However a complete set of high-resolution figures was submitted in parallel with the manuscript.

221: Why, in line 61 of Supplementary Methods, are t_IgH events listed as being filtered out due to low prevalence, when here in the main manuscript they are described as being merged together?

Thank you for your question regarding the discrepancy between line 61 of the Supplementary Methods and line 221 of the main manuscript.

We apologize for the confusion. The mention of t-IgH events being filtered out in the Supplementary Methods is indeed a typographical error. In fact, the list in the Supplementary Methods represents the events that were retained, not those that were filtered out.

In the main manuscript, we correctly state that t-IgH events were merged together. We corrected this error in the Supplementary Methods to ensure consistency and avoid any confusion.

247–254: Panels A and B should be described in the figure description. On line 253, the meaning of “resumed the clusters’ position” is not clear. How are the specific points depicted determined?

Thank you for your feedback regarding the description of Panels in line 253.

In response to your comments, we have made several improvements to the figure and its caption. The specific points depicted in the figure represent the centroids of the clusters, providing a summary representation of each cluster's position in the multi-dimensional scaling space. We have rearranged the panels to improve the interpretability of the figure and have revised the figure caption to reflect these changes.

The phrase "resumed the clusters’ position" has been clarified in the revised caption. It now explains that the positions of the centroids in the figure correspond to the positions of the clusters in the multi-dimensional scaling space, providing a visual summary of the clustering results.

263–266: Neither assumption of ancestry seems valid. A more prevalent genomic trait does not necessarily occur earlier than other traits—it may simply occur and reach high allele frequency (due to a

combination of high mutation rates and/or strong selection [13]) more often than less common variants. Similarly, association with a “simple genetic background” may indicate that the trait substantially contributes to MM without need for additional complex genomic alterations. However, both prevalence and lack of other explanatory events contribute to the argument for the importance of the variants, so a different framing in lieu of ancestry could be more convincing.

Thank you for your insightful feedback regarding our use of the term "ancestrality". We understand your concerns about the assumptions associated with this term and the potential for misinterpretation.

In response to your comment, we have decided to revise our manuscript to replace the term "ancestrality" with "driverness". This change reflects our intention to highlight the importance and potential functional impact of the genomic traits we identified, rather than making assumptions about their temporal occurrence in the evolution of multiple myeloma.

We believe that the term "driverness" more accurately conveys our findings and their implications, as it emphasizes the role of these genomic traits in driving the development and progression of multiple myeloma, rather than their order of occurrence.

286: Does “emersion” mean emergence here?

As mentioned in the previous response, the chapter has undergone revisions to enhance its significance. The term "emersion" has been removed from the chapter, resulting in a more refined presentation.

339–340: For clarity, please explain here why excluding t-IgH patients serves to confirm the DEG pattern. It becomes clear later in the manuscript.

An additional sentence explaining the reason of this exclusion has been added, as well as a reference supporting our explanation (line 352).

388: The notation is somewhat confusing: 1q&13+ means positive for gain of 1q and loss 13q, while 1q&13- means neutral copy number in both. Consider if there is another way to annotate the groups.

Thank you for your feedback regarding the notation used to denote the groups in our study. We understand your concern about potential confusion.

In our study, we used the notation "1q&13+" to denote a group that is positive for gain of 1q and loss of 13q, and "1q&13-" to denote a group that has neutral copy number for both. We believe this notation provides a concise and direct representation of the key genomic features defining each group.

The "+" and "-" symbols following the "&" are intended to quickly convey the status of the two chromosomes in question. The "+" indicates the presence of alterations (gain of 1q and loss of 13q), while the "-" indicates the absence of these alterations (neutral copy number).

We appreciate that this notation may require some initial explanation, but we believe that once understood, it provides a clear and efficient way to denote the groups.

1. Ortiz-Estévez M, Towfic F, Flynt E, Stong N, Jang IS, Wang K, et al. Integrative multi-omics identifies high risk multiple myeloma subgroup associated with significant DNA loss and dysregulated DNA repair and cell cycle pathways. *BMC Med. Genomics* 2021;14:295.
2. Hanamura I. Gain/Amplification of Chromosome Arm 1q21 in Multiple Myeloma. *Cancers* 2021;13:256.
3. Sklavenitis-Pistofidis R, Getz G, Ghobrial I, Papaioannou M. Multiple Myeloma With Amplification of Chr1q: Therapeutic Opportunity and Challenges. *Front. Oncol.* 2022;12:961421.
4. Flietner E, Yu M, Poudel G, Veltri AJ, Zhou Y, Rajagopalan A, et al. Molecular characterization stratifies VQ myeloma cells into two clusters with distinct risk signatures and drug responses. *Oncogene* 2023;42:1751–62.
5. Bolli N, Biancon G, Moarii M, Gimondi S, Li Y, de Philippis C, et al. Analysis of the genomic landscape of

- multiple myeloma highlights novel prognostic markers and disease subgroups. *Leukemia* 2018;32:2604–16.
6. Cardona-Benavides IJ, de Ramón C, Gutiérrez NC. Genetic Abnormalities in Multiple Myeloma: Prognostic and Therapeutic Implications. *Cells* [Internet] 2021;10. Available from: <http://dx.doi.org/10.3390/cells10020336>
 7. Maura F, Landgren O, Morgan GJ. Designing Evolutionary-based Interception Strategies to Block the Transition from Precursor Phases to Multiple Myeloma. *Clin. Cancer Res.* 2021;27:15–23.
 8. Dao P, Kim YA, Wojtowicz D, Madan S, Sharan R, Przytycka TM. BeWith: A Between-Within method to discover relationships between cancer modules via integrated analysis of mutual exclusivity, co-occurrence and functional interactions. *PLoS Comput. Biol.* 2017;13:e1005695.
 9. Kim YA, Cho DY, Dao P, Przytycka TM. MEMCover: integrated analysis of mutual exclusivity and functional network reveals dysregulated pathways across multiple cancer types. *Bioinformatics* 2015;31:i284–92.
 10. Castaneda O, Baz R. Multiple Myeloma Genomics - A Concise Review. *Acta Med. Acad.* 2019;48:57–67.
 11. Nishihori T, Shain K. Insights on Genomic and Molecular Alterations in Multiple Myeloma and Their Incorporation towards Risk-Adapted Treatment Strategy: Concise Clinical Review. *Int. J. Genomics Proteomics* 2017;2017:6934183.
 12. Klein MI, Cannataro VL, Townsend JP, Newman S, Stern DF, Zhao H. Identifying modules of cooperating cancer drivers. *Mol. Syst. Biol.* 2021;17:e9810.
 13. Cannataro VL, Gaffney SG, Townsend JP. Effect Sizes of Somatic Mutations in Cancer. *J. Natl. Cancer Inst.* 2018;110:1171–7.

Reviewer #3

(Remarks to the Author):

Reviewers' Comments:

Reviewer #1:

Remarks to the Author:

In this study the authors describe a new entity of MM using large genomic dataset and unsupervised method with the aim of defining a new feature of high risk. While the observation of the existence of this subgroup of patients in 25% of newly diagnosed MM patients with a unique genomic landscape is novel and important, the conclusion that co-occurrence of gain 1q and loss 13q correlates with high-risk is not fully supported by the data. Does co-occurrence of 1q gain & 13 loss able to capture high-risk patients who are "missed" or misclassified by the traditional scoring systems? Moreover, transcriptomic analysis is interesting, but the comparison shown in figure 6b is biased as they only compared gain 1q and loss 13q vs absence of both. 1q&13+ should be compared to all patients (1q&13- and 1q/13+), if the aim is to describe a transcriptional landscape specific to the co-occurrence of these 2 CA. Moreover, the analysis is not followed by any functional validation, and therefore does not provide any actionable therapeutic target that would have increased the significance to the paper. Other specific comments are below:

174-176: "Since rare genomic lesions gave a likely inferior contribution to MM genomic complexity, they were not included in this analysis, to avoid any excessive scattering of the matrix."

This statement is still confusing. Myeloma genomic complexity often refers to the genetic landscape in plasma cells. Some genomic lesions, even if rare, can create an important chromosomal instability and a complex karyotype.

198: correlation score between gain 1q and deletion 13q should be given since the following steps of the study focus on this co-occurrence.

293-298: "The co-occurrence of gain 1q and loss 13q [...] seemed to be a driver event in the progression."

This statement is not accurate because the DI score was calculated for gain 1q and del13q separately and not together.

525-526: "Ancestrality" was not corrected, as well as co-segregation which still appears across the manuscript.

Figures are still confusing. The colors of the 1q/13q do not match across the manuscript which make difficult the comprehension. Many typos are found in the text and in the figures.

Reviewer #2:

Remarks to the Author:

This reviewer previously stated that the clustering approach employed by the authors is not informative because the choice of clustering input (amp1q, presence of any odd-chromosome trisomy, del13q, tIgH) effectively pre-supposes the clusters. The revised manuscript does not substantively address this issue. Bolli 2018 (now cited in the revised manuscript) performed a more useful MM clustering analysis.

The manuscript argues for a need for a "univocal MM scoring system," the key claim being that patients with the combination amp1q/del13q have worse survival. If this finding is biologically valid, then the identification of this patient subgroup will be an important contribution. However, the manuscript does not present sufficient evidence that the survival differences are not due to chance or other explanatory factors. In the multivariate Cox modeling reported in Figure 8, the effects of amp1q/del13q alone on survival were no longer significant in the MM-BO data set, with worse outcomes instead found in the amp1q/del13q/t-IgH subgroup. As around 40–50% of amp1q/del13q patients in analyzed cohorts have t-IgH, it is critical to consider whether the reported high-risk group should be amp1q/del13q/t-IgH (or some other unconsidered classification) before suggesting, as the authors do in the abstract, the implementation of a "novel MM clinical stratification" based on amp1q/del13q alone.

Reviewer #3:

Remarks to the Author:

Reviewer #4:

Remarks to the Author:

The authors present the revised version of their elaborate work unveiling gain 1q in combination with loss of 13q a high risk constellation in MM. The previous concerns were adequately addressed and the quality of the manuscript has much improved. The science appears sound and the results are clinically important.

However, some concerns remain-

- While the authors elegantly show that 1q gain and 13q gain constellate a distinct genomic and transcriptomic group, it would be interesting to mention if the co-occurrence of both alterations has worse clinical outcome than 1q gain alone, which in itself is now seen a high risk feature. The authors only compare to a group consisting of either 1q gain or 13q loss, which might blur the effects of 1q alone. Furthermore, it's not clear how many patients in this group had which alteration. Please add this information and clarify

- Figures- the letters in the labels of all figures are way too small and difficult to read. Zooming in just makes the text go blurry. The dark colors in the Venn diagram in Figure 1 C make the letters illegible. Please adjust.

- Figure 5 has bad resolution and is not well legible

- The authors claim that gain 1q and loss 13q was an independent risk factor in multivariate analysis, however 1q&13q is not listed in the model in Figure 8. Please clarify.

- Line 451-454-> it is not quite clear what was done here. What is meant with complex genomic configuration? Was 1q&13q evaluated in combination with any of the MAF translocations? Please clarify and explain what is the rationale for that.

- Though the authors were asked to get rid the term of "ancestrality", the term driverness seems awkward and would be best replaced with something like "driver potential"

- o Eg. line 275 please change to, "we sought to measure their potential as oncogenic drivers"

- o Line 285- "Driver Index (DI)"

- o Line 291- Therefore, the higher the DI resulted, the more the genetic alteration was considered to be an oncogenic driver.

- o Line 294-, loss 13q and gain 1q were the top driver aberrations, ...

- The title seems a bit awkward and is somewhat non-telling. Please change to something like "Multi-

dimensional scaling techniques identify gain1q&loss13q co-occurrence as a high risk group in Multiple Myeloma with unique genomic and transcriptional features and adverse clinical outcome.”

Furthermore, there remain some issues with typos and language as following-

- Figure 4- change AI score to DI score

- Line 46 in the abstract “...highlighted a previously unrecognized patients’ unsupervised distribution in the low-dimensionality space.....”- it is unclear what this sentence means, please clarify/simplify

- Line 69-72- is also difficult to understand would also be better with some simplification. Are you saying discrete subgroups of patients could be defined by co-occurring, nonrandomly distributed events? Or conditional dependencies?

- Line 312- please change none to any (did not carry any of these..)

- Line 323-325- focal lesions is not the right term here, as it usually refers to anatomical lesions. Also, over-expressed does not seem to be the right term here as these are not transcriptomic analysis, but genomic. So, consider something like “several aberrations were enriched in 1q&13+ patients.

- Line 376- osteoclastogenesis process-> consider changing to osteoclast formation.

- Please check that all co-segregation is changed to co-occurrence (eg line 522 in discussion) and all “ancestrality” to either driver or driver potential (or similar), line 525 and 526 of discussion.

RESPONSES TO REVIEWER COMMENTS

Reviewer #1 (Remarks to the Author):

1. In this study the authors describe a new entity of MM using large genomic dataset and unsupervised method with the aim of defining a new feature of high risk. While the observation of the existence of this subgroup of patients in 25% of newly diagnosed MM patients with a unique genomic landscape is novel and important, the conclusion that co-occurrence of gain 1q and loss 13q correlates with high-risk is not fully supported by the data. Does co-occurrence of 1q gain & 13 loss able to capture high-risk patients who are “missed” or misclassified by the traditional scoring systems?

Thanks for your observations. We firstly would like to highlight that we ended up with the 1q&13 stratification starting from the analysis of MM genomic characteristics; therefore, our approach was strictly biological rather than statistical and a direct comparison with purely statistical risk-scoring systems may be not appropriate in the context of the present work (whereas it will be focused and elaborated in our next paper).

However, to answer to your comment, we would like to point out that in the multivariate model presented in Figure 8, both ISS3 (as defined by the conventional ISS scoring system) and 1q&13+ (as defined in the present paper) patients resulted at high risk of progression and death, suggesting that information provided both by ISS scoring system and 1q&13 classification are independent in defining risk, and not mutually exclusive.

From this result, it is clear that both classifiers might be useful in effectively describing high risk features, even though identifying different subgroups of patients at high risk.

The non-overlap between the two classifiers is further illustrated in the new Supplementary table 8 a,b. Here, it can be observed that patients carrying 1q&13+ are distributed across various ISS and R-ISS classes in a cross-sectional manner, and are even present in the low risk ISS1 and R-ISS I classes, demonstrating the added value of 1q&13 classifier in capturing high-risk patients, who might be misclassified by traditional scoring systems.

The table has been added as Supplementary Table S8, cited in the text in lines 446-450.

a

	Overall	ISS 1	ISS 2	ISS 3	non-classified	p	test
n	840	296	287	224	33		
MMrisk_class (%)						0.060	
1q&13-	336 (40.0)	135 (45.6)	109 (38.0)	79 (35.3)	13 (39.4)		
1q&13+	187 (22.3)	56 (18.9)	69 (24.0)	59 (26.3)	3 (9.1)		
1q/13	317 (37.7)	105 (35.5)	109 (38.0)	86 (38.4)	17 (51.5)		
MMrisk_class_t_CCND2 (%)						0.002	
1q&13-	336 (40.0)	135 (45.6)	109 (38.0)	79 (35.3)	13 (39.4)		
1q&13+_pure	112 (13.3)	37 (12.5)	42 (14.6)	32 (14.3)	1 (3.0)		
del 13q only	207 (24.6)	79 (26.7)	73 (25.4)	49 (21.9)	6 (18.2)		
gain 1q only	110 (13.1)	26 (8.8)	36 (12.5)	37 (16.5)	11 (33.3)		
t&1q&13+	75 (8.9)	19 (6.4)	27 (9.4)	27 (12.1)	2 (6.1)		

b

	Overall	R-ISS 1	R-ISS 2	R-ISS 3	non-classified	p	test
n	840	169	443	73	155		
MMrisk_class (%)						<0.001	
1q&13-	336 (40.0)	92 (54.4)	164 (37.0)	18 (24.7)	62 (40.0)		
1q&13+	187 (22.3)	22 (13.0)	106 (23.9)	29 (39.7)	30 (19.4)		
1q/13	317 (37.7)	55 (32.5)	173 (39.1)	26 (35.6)	63 (40.6)		
MMrisk_class_t_CCND2 (%)						<0.001	
1q&13-	336 (40.0)	92 (54.4)	164 (37.0)	18 (24.7)	62 (40.0)		
1q&13+_pure	112 (13.3)	22 (13.0)	62 (14.0)	9 (12.3)	19 (12.3)		
del 13q only	207 (24.6)	37 (21.9)	113 (25.5)	17 (23.3)	40 (25.8)		
gain 1q only	110 (13.1)	18 (10.7)	60 (13.5)	9 (12.3)	23 (14.8)		
t&1q&13+	75 (8.9)	0 (0.0)	44 (9.9)	20 (27.4)	11 (7.1)		

- Moreover, transcriptomic analysis is interesting, but the comparison shown in figure 6b is biased as they only compared gain 1q and loss 13q vs absence of both. 1q&13+ should be compared to all patients (1q&13- and 1q/13+), if the aim is to describe a transcriptional landscape specific to the co-occurrence of these 2 CA. Moreover, the analysis is not followed by any functional validation, and therefore does not provide any actionable therapeutic target that would have increased the significance to the paper.

Thanks for your comment, we acknowledge that the choice of differentially expressed genes (DEGs) comparisons among patients' sub-groups in the present study was not simple, as we chose to compare the 3 identified subgroups, instead of 1 versus the remaining 2. To this aim, we compared DEGs profiles between 2 groups at time (i.e. 1q&13+ vs. 1q&13-, 1q&13+ vs. 1q/13 and 1q&13- vs. 1q/13). As described in supplementary table S4, just one comparison (1q&13+ vs. 1q&13-) resulted in a list of significantly DEGs. Therefore, we guess that our analysis was not biased, since, in order to describe the expression profile derived from the co-occurrence of gain 1q and loss13q, we compared the gene expression profile of the patients' group carrying both these chromosomal aberrations with that of all the others patients' group in separate analyses, and we reported in fig.6b just the significant result. This procedure is described in lines 343-346. However, to further support our observation, according to the reviewer's request, we performed a new DEGs profile analysis, by comparing 1q&13+ patients' group vs all other patients (aggregation of 1q&13- and 1q/13 groups). As shown in the newly generated figures here below, the analysis confirmed the results reported in the paper, showing that the most significantly up and down regulated genes in 1q&13+ patients were *CCND2* and *CCND1*, respectively (even though with a slightly inferior fold change). Therefore, as this new analysis does not change the overall message of the paper's chapter, we would prefer to maintain the original comparisons, as described in the paper. However, if the reviewer judges this fundamental, we would include these additional data as supplementary figures.

3. Other specific comments are below:

174-176: "Since rare genomic lesions gave a likely inferior contribution to MM genomic complexity, they were not included in this analysis, to avoid any excessive scattering of the matrix." This statement is still confusing. Myeloma genomic complexity often refers to the genetic landscape in plasma cells. Some genomic lesions, even if rare, can create an important chromosomal instability and a complex karyotype.

Thank you for pointing out the ambiguity in lines 174-176. We would like to clarify that the rare genomic alterations were excluded from Figure S2 and Figure 2 just for a better visualization purpose. The primary objective of this specific figure was to represent the correlations among the most frequent CNAs in the overall CNAs' landscape. However, it is essential to highlight that rare genomic alterations were indeed retained in all analyses of the study. Specifically, we confirm that in the NMDS, aimed at studying the overall landscape's complexity, as well as in subsequent analyses (Figure 3, 4, and 5), all genomic variables were evaluated, including the rare ones.

We acknowledge that the phrasing reported in lines 174-176 might have been misleading; we therefore revised the text and the figure's legend to more accurately clarify our rationale (lines 173-175 and text describing Figure 2)

We appreciate your feedback and hope this clarification addresses your concerns.

198: correlation score between gain 1q and deletion 13q should be given since the following steps of the study focus on this co-occurrence.

Thanks for your suggestion, we agree that correlation score between gain 1q and loss 13q can add a valuable information to the reader and we added that score in the text (line 183-184).

293-298: "The co-occurrence of gain 1q and loss 13q [...] seemed to be a driver event in the progression." This statement is not accurate because the DI score was calculated for gain 1q and del13q separately and not together.

Thanks for your suggestion, we recognize that we used a confusing wording in this sentence. The text was changed from "co-occurrence" to "occurrence of both" in order to convey more accurately the intended message.

525-526: "Ancestrality" was not corrected, as well as co-segregation which still appears across the manuscript. Figures are still confusing. The colors of the 1q/13q do not match across the manuscript which make difficult the comprehension. Many typos are found in the text and in the figures.

We apologize for the oversight. We have now corrected the colors for 1q/13q to ensure consistency across the manuscript and have addressed the typos in both the text and the figures. Thank you for bringing this to our attention, and we appreciate your feedback.

Reviewer #2 (Remarks to the Author):

1. This reviewer previously stated that the clustering approach employed by the authors is not informative because the choice of clustering input (amp1q, presence of any odd-chromosome trisomy, del13q, tIgH) effectively pre-supposes the clusters. The revised manuscript does not substantively address this issue. Bolli 2018 (now cited in the revised manuscript) performed a more useful MM clustering analysis.

Thank you for your comment regarding the clustering approach. To address your concern, we would like to clarify that our clustering analysis pre-supposes just those clusters that have been known for a long time in the MM-related literature, specifically the well-known *Hyperdiploidy* and *t-IgH* clusters. In fact, the clustering analysis does not pre-suppose the new clusters, that have been indeed discovered in the present study and represent the main focus of our work, that is the co-occurrence of amp1q and del13 in a homogeneous cluster (1q&13+), clearly separated from the cluster lacking both amp1q and del13 (1q&13-).

From a methodological point of view, since all variables defining *Hyperdiploidy* are both strongly correlated to each other and redundant (as shown in Figure 2), we guess it would be appropriate to aggregate them, to avoid excessive multicollinearity effects that, in turn, could distort distances in the reduced-dimensionality space.

However, to avoid any doubt, we repeated the analysis by keeping separated all input variables and by not imposing any *a priori* structure to the data, similarly to the approach described in Bolli 2018. The results of this new analysis, which we attached here below, confirmed the strength of our findings: indeed, the 1q&13+ and 1q&13- clusters were both identified as well-separated clusters in the latent space.

We believe that our clustering input choice is justified by the acquired knowledge of MM genomic context, which represents the only “imposed structure”. Moreover, since results attained from the new analysis are overall in agreement with the core message of the paper, we feel confident by considering our starting hypothesis as not precluding the significance of the identified clusters.

We hope this addresses your concerns and provides further clarity on our methodology and findings.

2D dimensionality reduction with all separated variables

Hyperdiploidy - separated variables 2D NMDS

t IgH - separated variables 2D NMDS

Amp 1q - separated variables 2D NMDS

Del 13 - separated variables 2D NMDS

1q&13 classes - separated variables 2D NMDS

1q&13+ vs others - separated variables 2D NMDS

3D dimensionality reduction with all separated variables (as presented in figure 3)

2. The manuscript argues for a need for a “univocal MM scoring system,” the key claim being that patients with the combination amp1q/del13q have worse survival. If this finding is biologically valid, then the identification of this patient subgroup will be an important contribution. However, the manuscript does not present sufficient evidence that the survival differences are not due to chance or other explanatory factors. In the multivariate Cox modeling reported in Figure 8, the effects of amp1q/del13q alone on survival were no longer significant in the MM-BO data set, with worse outcomes instead found in the amp1q/del13q/t-IgH subgroup. As around 40–50% of amp1q/del13q patients in analyzed cohorts have t-IgH, it is critical to consider whether the reported high-risk group should be amp1q/del13q/t-IgH (or some other unconsidered classification) before suggesting, as the authors do in the abstract, the implementation of a “novel MM clinical stratification” based on amp1q/del13q alone.

Thank you for your comment: we guess the overall clinical message has been over-estimated and we would like to better explain the rationale driving the clinical analysis, that we will also try to implement within the paper.

In fact, the main objective of the present paper has been the biological stratification of Multiple Myeloma patients, based on the genomic profile of CNAs and FISH-detected t-IgH; these alterations are known to highly contribute to MM genomic heterogeneity.

Once defined the novel patients’ stratification, we decided to deepen our finding by exploring the clinical outcome of the three sub-groups of patients, describing their progression-free and overall survival. We highlighted that patients 1q&13+, possibly due to the observed deregulated expression of critical pathways, have worse prognosis, as compared to the others. This clinical model (that we would like to name “model 1”) is not based on statistical considerations, and do not aim at the identification of a scoring system.

Subsequently, since 1q&13, t(4;14) and MAF-translocations, all have been shown to significantly deregulate *CCDN2* expression (as shown in Figure 5), we decided to explore also the prognostic impact of this novel genomic combination, carried by a small sub-group of patients and named t&1q&13, and we defined this second model “model 2”. From a biological point of view, this genomic combination - again not statistically-driven - includes genomic aberrations that cause a cell cycle de-regulation and an incremented proliferation, as already reported in literature (see below).

We observed that the co-occurrence of these genomic aberrations, even though rare, confers a significantly high risk of progression and death to patients and therefore might be considered a clinical biomarker. Notably, neither this clinical model (model 2) is based on statistical consideration nor do aim at the identification of a scoring system; we will address this issue in a future paper, by formally comparing the prognostic impact of our biologically-defined scoring system to the conventionally employed (e.g R-ISS and R2-ISS), which is not the aim of the present paper.

We acknowledge that the rationale driving the overall clinical analyses has not been clearly-enough explained in the Results and in the Discussion, that we modified accordingly (lines 451-465 of Results and lines 478-480, explaining Figure 8; lines 602-639 of Discussion).

- **Del 13: RB1 inactivation:** “widely known tumor suppressor gene, it regulates cell cycle and prevents excessive cell growth.” Ref: Chinnam, Meenalakshmi, and David W. Goodrich. "RB1, development, and cancer." *Current topics in developmental biology* 94 (2011): 129-169. <https://doi.org/10.1016/B978-0-12-380916-2.00005-X>
- **Amp 1q: CKS1B overexpression:** “activates cyclin-dependent kinases, main role in cell-cycle and proliferation.” Ref: Zhan, Fenghuang, et al. "CKS1B, overexpressed in aggressive disease, regulates multiple myeloma growth and survival through SKP2-and p27Kip1-dependent and-independent mechanisms." *Blood, The Journal of the American Society of Hematology* 109.11 (2007): 4995-5001. <https://doi.org/10.1182/blood-2006-07-038703>
- **t(4;14): FGFR3 overexpression:** “promotes cell cycle progression.” Ref: Brito, Jose LR, et al. "MMSET deregulation affects cell cycle progression and adhesion regulons in t (4; 14) myeloma plasma cells." *haematologica* 94.1 (2009): 78. <https://doi.org/10.3324/haematol.13426>

- **MAF traslocations: c-maf overexpression:** “c-maf transforms plasma cells by stimulating cell cycle progression.” Ref: Hurt, Elaine M., et al. "Overexpression of c-maf is a frequent oncogenic event in multiple myeloma that promotes proliferation and pathological interactions with bone marrow stroma." Cancer cell 5.2 (2004): 191-199. [https://doi.org/10.1016/S1535-6108\(04\)00019-4](https://doi.org/10.1016/S1535-6108(04)00019-4)

Reviewer #3 (Remarks to the Author):

Reviewer #4, new reviewer with expertise in MM biology and genomics (Remarks to the Author):

The authors present the revised version of their elaborate work unveiling gain 1q in combination with loss of 13q a high-risk constellation in MM. The previous concerns were adequately addressed and the quality of the manuscript has much improved. The science appears sound and the results are clinically important. However, some concerns remain.

1. While the authors elegantly show that 1q gain and 13q gain constellate a distinct genomic and transcriptomic group, it would be interesting to mention if the co-occurrence of both alterations has worse clinical outcome than 1q gain alone, which in itself is now seen a high risk feature. The authors only compare to a group consisting of either 1q gain or 13 q loss, which might blur the effects of 1q alone. Furthermore, it's not clear how many patients in this group had which alteration. Please add this information and clarify

Thank you for your observation. We would like to highlight that the survival effect of the "1q gain alone" group, is shown both by the multivariate analysis in Figure 8 and by the Kaplan-Meier curves in Supplementary Figure 6. Indeed, in these analyses we considered the groups "only 1q" and "only 13", and not the combined "1q/13" group. Notably, the analysis showed that the "1q gain alone" group's impact on survival is no longer significant when included in a multivariate analysis along with that of "1q&13+" group. Therefore, we guess that the effect of the "1q gain alone" presence has been taken into account and was not diluted in the "1q/13" group in the clinical analysis.

We concur that the text does not convey how many patients have either 1q gain or 13 loss alone, as this information is only present in the forest plot in Figure 8. To this aim, we added detailed information on this aspect in the section introducing the "1q&13 classification" (lines 309-310).

We hope this provides clarity on our methodology and findings, and we appreciate your feedback.

2. Figures- the letters in the labels of all figures are way too small and difficult to read. Zooming in just makes the text go blurry. The dark colors in the Venn diagram in Figure 1 C make the letters illegible. Please adjust.
3. Figure 5 has bad resolution and is not well legible

We acknowledge the concerns regarding the legibility of the labels and the resolution of figures (issues 2 and 3). However, the figures provided in the Word document are of reduced resolution for file size considerations. However, please note that we have provided high-resolution versions of all figures in separate PDF files to ensure clarity and legibility. We will ensure that these high-resolution figures are used in the final version of the manuscript.

4. The authors claim that gain 1q and loss 13q was an independent risk factor in multivariate analysis, however 1q&13q is not listed in the model in Figure 8. Please clarify.

We would like to clarify that the "1q&13+" is indeed included in the model 1 presented in Figure 8. It is part of the "MMrisk_allclass" variable, along with other categories of the "1q&13" classification, namely "1q&13-", "gain 1q only", and "del 13q only". Our claim is supported by the fact that the "1q&13+" category is found to be significant in all models, both for OS and PFS, and this holds true for both the COMPASS and BO datasets, as shown in forest plots of Figure 8.

5. Line 451-454-> it is not quite clear what was done here. What is meant with complex genomic configuration? Was 1q&13q evaluated in combination with any of the MAF translocations? Please clarify and explain what is the rationale for that.

We apologize for the confusion, we recognize that we used a bad wording. The text was changed accordingly from "complex genomic configuration" to "genomic combination". In addition, we added a substantial part in the discussion in order to clarify our rationale and our choice of evaluating the t&1q&13 group (1q&13q in combination with any of the translocations deregulating CCND2 – ie. t(4;14), t(14;16) and t(14;20)) as a biological risk-factor (Lines 451-454).

6. Though the authors were asked to get rid the term of "ancestrality", the term driverness seems awkward and would be best replaced with something like "driver potential"
 - o Eg. line 275 please change to, "we sought to measure their potential as oncogenic drivers"
 - o Line 285- "Driver Index (DI)"
 - o Line 291- Therefore, the higher the DI resulted, the more the genetic alteration was considered to be an oncogenic driver.
 - o Line 294-, loss 13q and gain 1q were the top driver aberrations, ...

We apologize for any confusion caused by our terminology. We understand the challenge in finding an objective definition for a term associated with the concept it represents. We opted for the term "driverness" as it has been used in several papers, which we cite below, and the conveyed concept seems similar to our intent. We favor this term as we believe it is straightforward and has been commonly used in the literature. We also recognize the term could seem awkward if not introduced appropriately, so we added the line you suggested to better clarify this term meaning (lines 275-276).

- Bailey, Matthew H., et al. "Comprehensive characterization of cancer driver genes and mutations." *Cell* 173.2 (2018): 371-385. <https://www.ncbi.nlm.nih.gov/pmc/articles/PMC6029450/>
- Petrov, Iurii, and Andrey Alexeyenko. "Individualized discovery of rare cancer drivers in global network context." *Elife* 11 (2022): e74010. <https://www.ncbi.nlm.nih.gov/pmc/articles/PMC9159755/>
- Mukherjee, Sumit, et al. "Identifying and ranking potential driver genes of Alzheimer's disease using multiview evidence aggregation." *Bioinformatics* 35.14 (2019): i568-i576. <https://www.ncbi.nlm.nih.gov/pmc/articles/PMC6612835/>

7. The title seems a bit awkward and is somewhat non-telling. Please change to something like "Multi-dimensional scaling techniques identify gain1q&loss13q co-occurrence as a high risk group in Multiple Myeloma with unique genomic and transcriptional features and adverse clinical outcome."

Thank you for your suggestion regarding the title. Based on your feedback, we have revised the title to better reflect the message that gain1q&loss13q represents a group with an adverse clinical outcome. However, we found the proposed title to be quite lengthy, so we opted not to include the term "high risk" as it seemed redundant. We appreciate your input and have made the necessary adjustments.

Furthermore, there remain some issues with typos and language as following-

- Figure 4- change AI score to DI score

Corrected.

- Line 46 in the abstract "...highlighted a previously unrecognized patients' unsupervised distribution in the low-dimensionality space....."- it is unclear what this sentence means, please clarify/simplify

We simplified the sentence by moving the word "unsupervised" and changing the word "distribution" with "cluster".

- Line 69-72- is also difficult to understand would also be better with some simplification. Are you saying discrete subgroups of patients could be defined by co-operating, nonrandomly distributed events? Or conditional dependencies?

We are actually asserting both statements. We do not see these two concepts as separated, but connected by a logical thread: we claim that nonrandomly distributed co-operating events, which are logically connected through a conditional dependence relationship leading to the onset of cancer, can delineate discrete subgroups of patients. We therefore would like to maintain the sentence, as it conveys our intended message.

- Line 312- please change none to any (did not carry any of these..)

Done, thanks for your suggestion.

- Line 323-325- focal lesions is not the right term here, as it usually refers to anatomical lesions. Also, over-expressed does not seem to be the right term here as these are not transcriptomic analysis, but genomic. So, consider something like "several aberrations were enriched in 1q&13+ patients.

We changed our wording accordingly to your suggestion, apologies for the confusion.

- Line 376- osteoclastogenesis process-> consider changing to osteoclast formation.

Done, thanks for your suggestion. We also think that osteoclast formation is an easier but equivalent term.

- Please check that all co-segregation is changed to co-occurrence (eg line 522 in discussion) and all "ancestrality" to either driver or driver potential (or similar), line 525 and 526 of discussion.

Done.

Reviewers' Comments:

Reviewer #1:

Remarks to the Author:

The authors have addressed majority of the concerns. The study is relevant for the myeloma's field, as it highlights the presence of a group of myeloma patients characterized by co-occurrence of 1q gain and 13 loss with distinct transcriptomic profiles and unfavorable clinical outcome.

Reviewer #2:

Remarks to the Author:

This revision does not adequately address the issues that I raised in the previous review. The clustering analysis, based on a handful of features (amp1q, presence of any odd-chromosome trisomy, del13q, tIgH), does not produce useful insights, as it is to be expected that the resultant clusters consist of patients with various simple combinations of those features.

I am concerned that the authors have expressed a lack of care over whether the adverse clinical outcomes associated with 1q_amp/13_del are biologically valid findings. The rebuttal stated that a future paper will "formally" assess clinical risk. If there isn't confidence in the clinical relevance of the novel MM patient clustering, then publication seems premature.

Relatedly, I am perplexed by the following sentence that has been added to the manuscript: "1q&13+ patients were distributed across all ISS (and R-ISS for CpMMpass [sic]) classes and were even present in the low risk ISS1 (and R-ISS I) class, highlighting the added value of 1q&13 classifier in capturing high-risk patients, who might be misclassified by traditional scoring systems." Without the proper statistical analysis that the authors declined to perform, I do not understand how a feature that is present in both high- and low-risk patients could usefully contribute to identification of high-risk patients.

Reviewer #3:

Remarks to the Author:

Reviewer #4:

Remarks to the Author:

The authors have done a good job in addressing the comments and revising the manuscript. The text seems more scientific sound with improved clarification and language.

Please note that during the revision process, there is a duplicate sentence in lines 459-462. Though improved, the reviewer would still strongly recommend to enlarge the letters in most figures/tables and legends, they remain very small.

Response to Referees Letter – Version B – 10/19/2023

REVIEWERS' COMMENTS

Reviewer #1 (Remarks to the Author):

The authors have addressed majority of the concerns. The study is relevant for the myeloma's field, as it highlights the presence of a group of myeloma patients characterized by co-occurrence of 1q gain and 13 loss with distinct transcriptomic profiles and unfavorable clinical outcome.

Reviewer #2 (Remarks to the Author):

This revision does not adequately address the issues that I raised in the previous review. The clustering analysis, based on a handful of features (amp1q, presence of any odd-chromosome trisomy, del13q, tIgH), does not produce useful insights, as it is to be expected that the resultant clusters consist of patients with various simple combinations of those features.

Clustering analysis has been performed starting from *all* CNAs and IgH translocations (overall 67 features, see line 758 of the paper) observed in the genomes of MM patients, and not just by using a handful of features, as mentioned by the reviewer. This is clearly stated in lines 229-230 of the paper. Therefore, the clusters resulting from the analysis are not trivial, as they specifically highlight the presence of not previously observed clusters consisting either in the co-occurrence or co-absence of both 1q amplification and 13q deletion. This result could not be up-front expected, as it emerged, as novel discovery, from our unsupervised clustering analysis that included all the CNAs and t-IgH genomic variables. The list of variables included in clustering analysis is mentioned in lines 757-759 and described in Supplementary material.

I am concerned that the authors have expressed a lack of care over whether the adverse clinical outcomes associated with 1q_amp/13_del are biologically valid findings. The rebuttal stated that a future paper will “formally” assess clinical risk. If there isn’t confidence in the clinical relevance of the novel MM patient clustering, then publication seems premature.

We would like to state that we are fully confident in the reported results, as in multivariate Cox model 1 and 2 we included all co-variates that could possibly explain the validity of the reported adverse clinical outcome associated to both 1q&13 and t&1q&13 classifications.

We also would like to underline that the building of a clinical risk scoring system, including the new reported features, requires a fully statistical approach, that was not the object of the present paper. Therefore, in the previous rebuttal letter, we anticipated that we will (actually we are) address this topic in a future paper, with a purely statistical (and not biological) approach.

In fact, in the letter we stated “Notably, neither this clinical model (model 2) is based on statistical consideration nor do aim at the identification of a scoring system; we will address this issue in a future paper, by formally comparing the prognostic impact of our biologically-defined scoring system to the conventionally employed (e.g R-ISS and R2-ISS), which is not the aim of the present paper.”, meaning that aim of a future paper will be the definition of a *scoring system*, and not the *assessment of the clinical risk*, which has been already dissected in the present paper, with the actually most commonly employed statistical tools and procedures.

Relatedly, I am perplexed by the following sentence that has been added to the manuscript: “1q&13+ patients were distributed across all ISS (and R-ISS for CpMMpass [sic]) classes and were even present in the low risk ISS1 (and R-ISS I) class, highlighting the added value of 1q&13 classifier in capturing high-risk patients, who might be misclassified by traditional scoring systems.” Without the proper statistical analysis

that the authors declined to perform, I do not understand how a feature that is present in both high- and low-risk patients could usefully contribute to identification of high-risk patients.

We would like to reject the statement that we “declined” to perform a proper statistical analysis, since on the contrary we carefully addressed the raised issues.

In fact, in the multivariate Cox models, we did consider ISS along with 1q&13+ feature; in this way we jointly considered the hazard risk independently associated to both high and low risk classes, as defined by ISS scoring system, along with 1q&13 classification (model 1) and with t&1q&13 classification (model 2). Please note that since R-ISS data were not available for BO datasets, it has not been possible to perform the same comparative homogeneous analysis in both dataset, which we might acknowledge could be considered a limit of the study that, however is not possible to overcome. The claim of “miss-classification” should be considered relative to any given scoring system used as reference. Therefore, the identification of high-risk patients is still a huge open question in MM community, with multiple scoring systems available employed and no consensus reached so far; in fact, it is considered an unmet clinical need.

In this context, our results are intended to contribute to shed light to this critical clinical topic, by adding several biological insights; obviously the univocal definition of “high-risk” could not be definitively address in the present paper, even though we are convinced that by adding biological insights, we might contribute to better explain this adverse clinical behavior.

Reviewer #3 (Remarks to the Author):

Reviewer #4 (Remarks to the Author):

The authors have done a good job in addressing the comments and revising the manuscript. The text seems more scientific sound with improved clarification and language.

Please note that during the revision process, there is a duplicate sentence in lines 459-462. Though improved, the reviewer would still strongly recommend to enlarge the letters in most figures/tables and legends, they remain very small.